# Control of parallel hippocampal output pathways by amygdalar long-range inhibition

**Rawan AlSubaie, Ryan WS Wee, Anne Ritoux, Karyna Mishchanchuk, Jessica Passlack, Daniel Regester, Andrew F MacAskill\***

Department of Neuroscience, Physiology and Pharmacology, University College London, London, United Kingdom

**Abstract** Projections from the basal amygdala (BA) to the ventral hippocampus (vH) are proposed to provide information about the rewarding or threatening nature of learned associations to support appropriate goal-directed and anxiety-like behaviour. Such behaviour occurs via the differential activity of multiple, parallel populations of pyramidal neurons in vH that project to distinct downstream targets, but the nature of BA input and how it connects with these populations is unclear. Using channelrhodopsin-2-assisted circuit mapping in mice, we show that BA input to vH consists of both excitatory and inhibitory projections. Excitatory input specifically targets BA- and nucleus accumbens-projecting vH neurons and avoids prefrontal cortex-projecting vH neurons, while inhibitory input preferentially targets BA-projecting neurons. Through this specific connectivity, BA inhibitory projections gate place-value associations by controlling the activity of nucleus accumbens-projecting vH neurons. Our results define a parallel excitatory and inhibitory projection from BA to vH that can support goal-directed behaviour.

## Editor's evaluation

This manuscript represents an important piece of work that defines the cellular basis of hippocampal-amygdala functional connectivity in rodents.

**\*For correspondence:**
a.macaskill@ucl.ac.uk

## Introduction

The hippocampus is key for episodic memory, learning and spatial navigation, as well as motivation, affect and anxiety (*Gray and McNaughton, 2003*; *O'Keefe and Nadel, 1978*; *Strange et al., 2014*; *Wikenheiser and Schoenbaum, 2016*). At almost every level of investigation – from gene expression, to afferent and efferent connectivity, and behavioural function – the hippocampus is organised as a gradient along the dorsal to ventral (posterior to anterior in humans) axis (*Fanselow and Dong, 2010*; *Strange et al., 2014*). Within this axis the most dorsal portion is proposed to be involved in learning and utilising fine-grained spatial and temporal structure, whereas the most ventral pole is thought to be involved in affect and motivation, and has a key role in value-based and reward-driven decision-making and anxiety-like calculations (*Fanselow and Dong, 2010*; *Strange et al., 2014*).

A distinguishing factor that separates the ventral from the dorsal hippocampus is dense input from the corticobasolateral nuclear complex of the amygdala (basal amygdala [BA]; *McDonald and Mott, 2017*; *Strange et al., 2014*). The BA comprises a diverse set of nuclei including the basolateral amygdala (BLA), the basomedial amygdala (BMA), the medial amygdala (MEA) and cortical amygdala, each of which sends projections to ventral hippocampus (vH) (*McDonald and Mott, 2017*; *Petrovich et al., 2001*; *Strange et al., 2014*). These nuclei, and their projections to vH, are thought to be crucial

for the learning of reward- and threat-associated cues, and for the generation of appropriate goal-directed and anxiety-like behaviour (*Beyeler et al., 2018*; *Beyeler et al., 2016*; *Felix-Ortiz et al., 2013*; *Felix-Ortiz and Tye, 2014*; *Hitchcott and Phillips, 1997*; *McHugh et al., 2004*; *Pi et al., 2020*; *Richardson et al., 2004*; *Selden et al., 1991*; *Sheth et al., 2008*; *Yang and Wang, 2017*). Thus, it is commonly assumed that powerful and specific synaptic connectivity between these two structures is crucial for the maintenance of such behaviours. However, there is limited information describing the organisation of such functional connectivity between amygdala input and neurons in vH (*Felix-Ortiz et al., 2013*; *Pi et al., 2020*).

This lack of understanding is compounded by the fact that the vH, in particular its output structure the ventral CA1 and subiculum – where the majority of BA input is found – is organised as a parallel circuit, such that the majority of neurons project to only one downstream area (*Gergues et al., 2020*; *Naber and Witter, 1998*; *Wee and MacAskill, 2020*). Thus, while vH has powerful connections to the nucleus accumbens (NAc), the prefrontal cortex (PFC) and back to the BA, each of these projections arises from a distinct population of neurons. Importantly each of these projection populations is increasingly shown to underlie unique behavioural functions (*Adhikari et al., 2010*; *Jimenez et al., 2018*; *LeGates et al., 2018*; *Sanchez-Bellot and MacAskill, 2021*). For example, vH$^{NAc}$ neuron activity is high during motivated behaviour and around rewarded locations (*Ciocchi et al., 2015*; *Okuyama et al., 2016*; *Reed et al., 2018*), is necessary for place-value associations (*LeGates et al., 2018*; *Trouche et al., 2019*) and can promote spatial and instrumental reinforcement (*Britt et al., 2012*; *LeGates et al., 2018*). In contrast, vH$^{PFC}$ activity is proposed to support the resolution of approach avoidance conflict and contribute to spatial working memory (*Padilla-Coreano et al., 2016*; *Sanchez-Bellot and MacAskill, 2021*; *Spellman et al., 2015*), while vH$^{BA}$ activity is proposed to support contextual learning (*Jimenez et al., 2018*). However, it remains unclear how the activity of these distinct populations in vH is differentially controlled to promote these functions. We reasoned that a means for this control would be projection-specific innervation from BA.

The circuit organisation of the nuclei in the BA is similar to classic cortical circuitry – with the majority of neurons classed as either excitatory pyramidal neurons or local inhibitory interneurons (*McDonald and Mott, 2017*). However, there is also evidence suggesting the presence of long-range inhibitory projection neurons throughout BA (*Dedic et al., 2018*; *McDonald et al., 2012*; *McDonald and Zaric, 2015*; *Seo et al., 2016*). Similar inhibitory projections from cortex are hypothesised to have a crucial regulatory role in modulating hippocampal circuit function (*Basu et al., 2016*; *Melzer et al., 2012*), but the connectivity and function of BA long-range inhibitory input in vH has never been directly investigated.

In this study, we used a combination of retrograde tracing, electrophysiology and channelrhodopsin-2-assisted circuit mapping to show that BA provides both excitatory and direct inhibitory input to distinct projection populations within vH. We show that excitatory projections uniquely target vH neurons that project to NAc and back to the BA, and do not connect with neurons that project to PFC. In contrast, long-range inhibitory input preferentially targets BA-projecting vH neurons. Next, using a simple network model constrained by our electrophysiology recordings, we predicted that the ability of BA input to drive motivation- and value-promoting vH projections to NAc was dependent on the co-activation of both excitatory and inhibitory input from BA. Finally, we confirmed these predictions using in vivo optogenetics and genetically targeted pharmacology to show that long-range inhibition is required for the generation of spatial place preference. Together, our results outline a novel inhibitory projection from amygdala to vH that defines the activity of vH output neurons and is able to control hippocampal output to promote the formation of spatial place preference.

## Results

### BA input into vH is both excitatory and inhibitory

While the majority of investigation of BA-vH connectivity is focussed on projections specifically from the BLA, it is known that multiple BA nuclei project to vH (*McDonald and Mott, 2017*). Therefore, we first determined the spatial distribution of neurons in BA that send input into vH by injecting a fluorescently conjugated cholera toxin beta subunit (CTXβ) into the ventral part of the hippocampus (*Figure 1A*). CTXβ is taken up by presynaptic terminals at the injection site and retrogradely transported to label the soma of afferent neurons. After 2 weeks, we serially sectioned labelled brains and

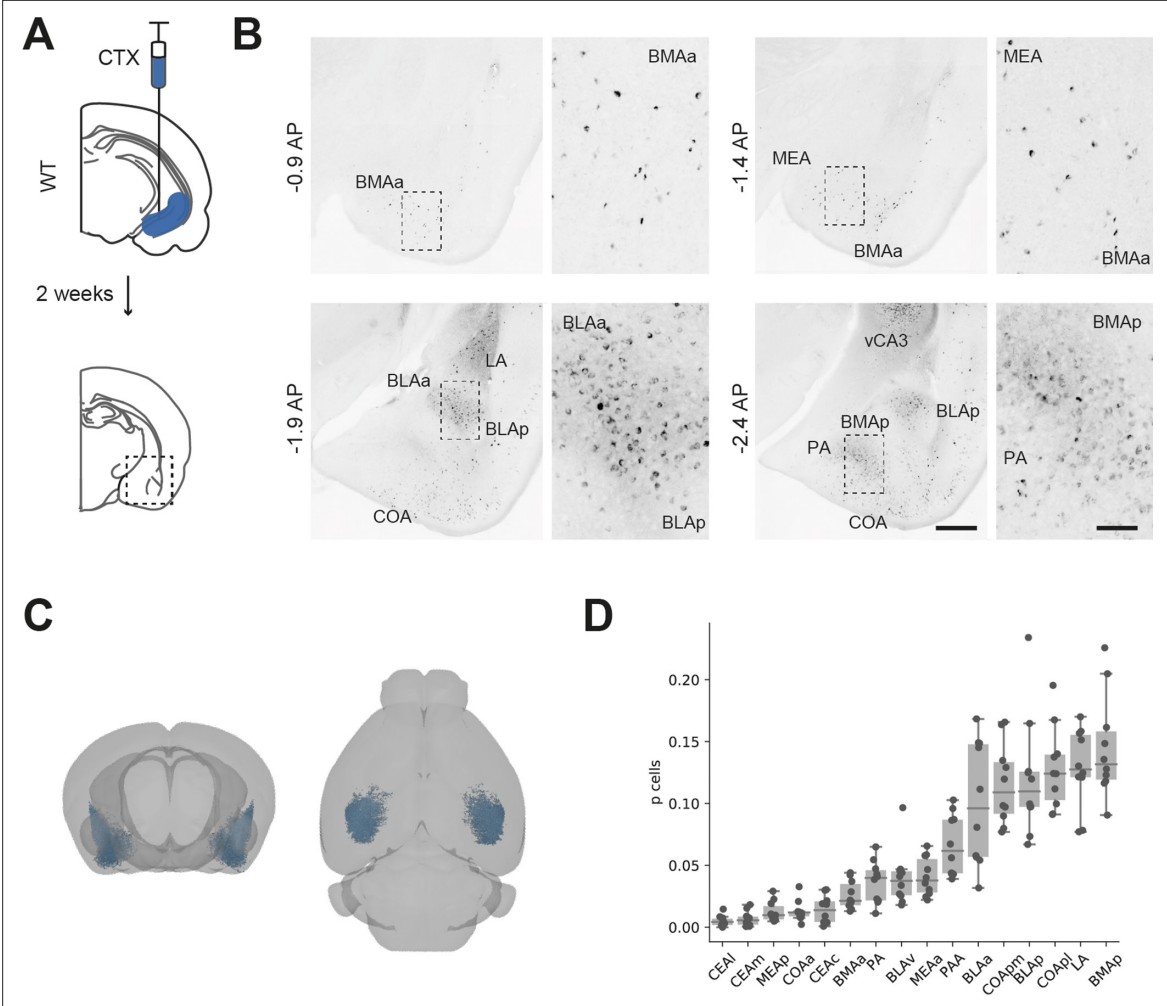

**Figure 1.** Distribution of basal amygdala (BA) input to ventral hippocampus (vH). (**A**) Schematic of experiment. CTXβ was injected into vH, 2 weeks later coronal slices of BA were examined for retrogradely labelled neurons. (**B**) Example slices showing widespread labelling throughout numerous BA nuclei. Scale bar = 500 μm, 100 μm (zoom). Images are stitched from tiled scans. (**C**) Whole-brain distribution of labelled BA neurons. (**D**) Summary showing proportion of labelled BA cells in each nuclei. CEA, central amygdala; MEA, medial amygdala; COA, cortical amygdala; BMA, basomedial amygdala; PA, posterior amygdala; BLA, basolateral amygdala; LA, lateral amygdala; l, lateral; m, medial; a, anterior; p, posterior.

The online version of this article includes the following figure supplement(s) for figure 1:

**Source data 1.** Source data for *Figure 1*.

mapped labelled cell locations to the Allen Brain Atlas (ABA) (*Fürth et al., 2018*; *Wee and MacAskill, 2020*). We found that neurons sending input to vH were widely dispersed throughout the entire BA, including in BLA, BMA and MEA, as well as in more cortical amygdala areas (*Figure 1B–D*, *McDonald and Mott, 2017*; *Strange et al., 2014*). Overall, this experiment confirmed that there is large input from disperse BA nuclei to vH, focussed around the posterior BMA and BLA.

We next tested whether BA input to vH may be both excitatory and inhibitory (*McDonald and Mott, 2017*). We repeated our experiment using a vGAT-cre::dtomato reporter mouse. In this experiment, CTXβ-labelled neurons in BA could be distinguished as either GABAergic (vGAT+) or putatively excitatory (vGAT-) based on fluorescence colocalisation. Using this approach, we found that a small but consistent proportion of BA neurons (3.7% of CTXβ-labelled neurons) that projected to vH were GABAergic (*Figure 2A*, *Figure 2—figure supplement 1A and B*). Using whole-brain registration as before, we found that inhibitory projection neurons were intermingled with excitatory projection neurons, such that there was no obvious anatomical separation between inhibitory and classic excitatory projections. Supporting this, both were found in consistent proportions (~4% of labelled neurons) throughout each nucleus in BA and across all three anatomical axes (*Figure 2—figure*

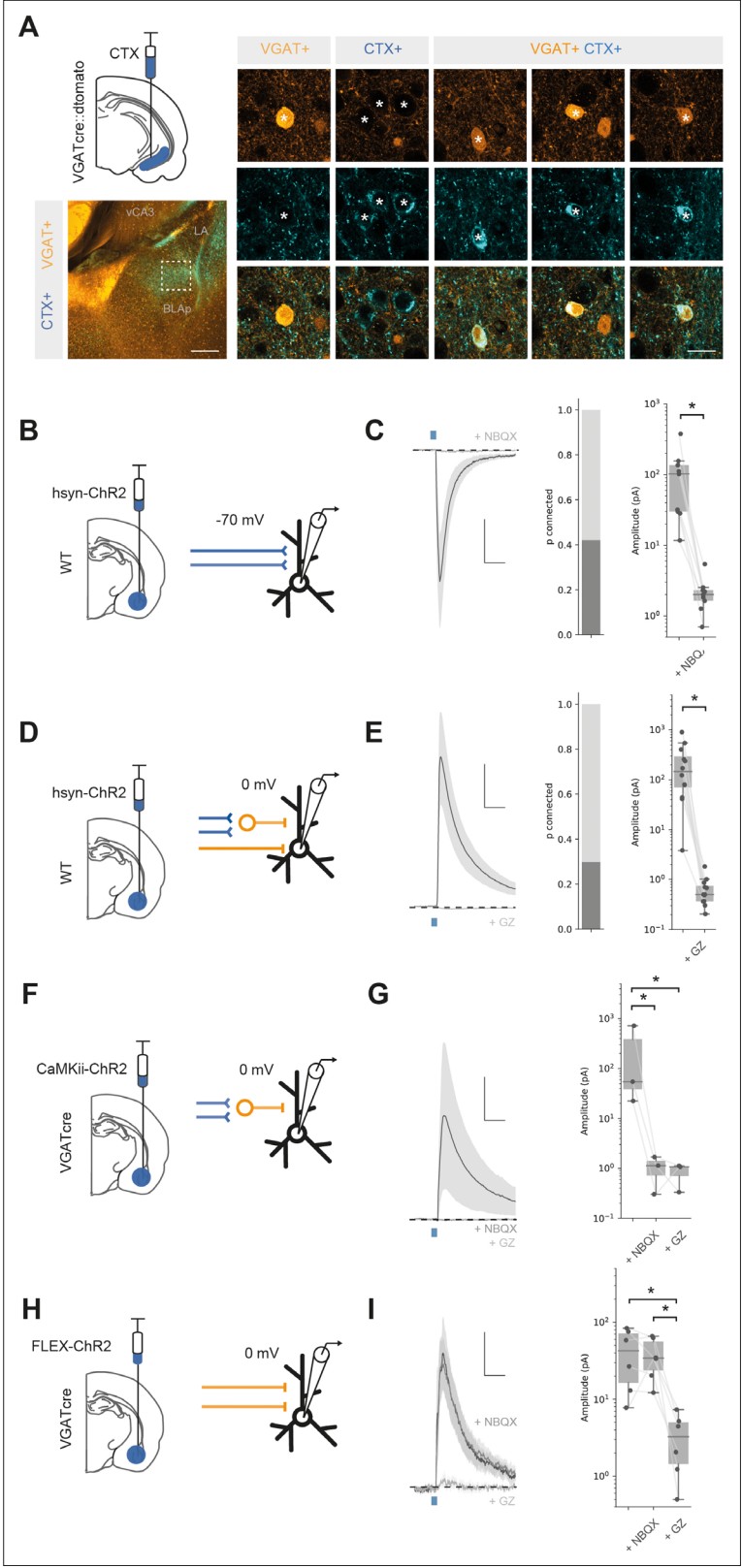

**Figure 2.** Basal amygdala (BA) input to ventral hippocampus (vH) is both excitatory and inhibitory. (**A**) CTXβ injection in vH in a vGAT::cre::dtomato mouse line reveals inhibitory neurons (vGAT+), putative excitatory neurons that project to vH (CTX+) and inhibitory neurons that project to vH (vGAT+ CTX+). Example neurons from boxed region on left. Scale bar = 300 µm (left), 20 µm (right). (**B**) Schematic showing experimental setup. ChR2 was

*Figure 2 continued on next page*

*Figure 2 continued*

expressed using the pan-neuronal synapsin promoter using an adeno-associated virus (AAV) injection in BA. After allowing for expression, whole-cell recordings were performed in voltage clamp at – 70 mV in vH. (**C**) Brief pulses of blue light evoke excitatory currents that are blocked by the AMPA receptor antagonist NBQX. Left: average current trace pre- and post-NBQX. Middle: proportion of recorded cells connected (with time-locked response to light). Right: amplitude before and after NBQX. Note log scale. NBQX blocks excitatory currents evoked by BA input. Scale bar = 50 pA, 10 ms. (**D, E**) As (**B, C**) but for voltage clamp at 0 mV before and after the GABA receptor antagonist gabazine. Gabazine blocks inhibitory currents evoked by BA input. Scale bar = 50 pA, 10 ms. (**F**) Feedforward inhibition isolated using ChR2 expression under the CaMKii promoter. (**G**) Brief pulses of blue light evoked inhibitory currents at 0 mV that are blocked by the AMPA receptor antagonist NBQX. Left: average current trace pre- and post-NBQX and GZ. Right: amplitude before and after NBQX and GZ. Note log scale. NBQX blocks inhibitory currents evoked by CaMKii BA input, indicating it is solely feedforward. Scale bar = 50 pA, 10 ms. (**H, I**) As for (**F, G**) but direct inhibitory input isolated using ChR2 expression only in vGAT + BA neurons. NBQX has no effect on direct inhibitory connection, while it is blocked by GZ, indicating that it is a direct, long-range inhibitory connection. Scale bar = 15 pA, 10 ms.

The online version of this article includes the following figure supplement(s) for figure 2:

**Source data 1.** Source data for *Figure 2*.

**Figure supplement 1.** Feedforward and direct inhibitory input from basal amygdala (BA) to ventral hippocampus (vH).

**Figure supplement 2.** Somatostatin-positive neurons project from basal amygdala (BA) to ventral hippocampus (vH).

**Figure supplement 3.** Both CaMKii+ and VGAT+ neurons project from basal amygdala (BA) to ventral hippocampus (vH).

**Figure supplement 4.** Confirmation of synapsin+ and VGAT+ projection from basal amygdala (BA) to ventral hippocampus (vH) from Allen Brain Connectivity Atlas.

**Figure supplement 5.** Example injection sites for physiology experiments.

*supplement 1A–C*). Previous studies have suggested that long-range inhibitory input in vH arises from somatostatin-positive neurons (*McDonald and Mott, 2017*). Therefore, we repeated our CTXβ experiment and performed immunostaining against somatostatin. Consistent with previous results, we found that a proportion of CTX+ BA neurons projecting to vH were also somatostatin positive (*Figure 2—figure supplement 2*). Thus, in addition to the classically described excitatory projection from BA to vH, there is a parallel inhibitory projection arising from GABAergic neurons from across the BA, and these neurons are likely to express the peptide somatostatin.

We next investigated if these projections made functional connections onto vH pyramidal neurons. To recruit both excitatory and inhibitory projections from BA, we used channelrhodopsin-assisted circuit mapping (CRACM). We expressed ChR2 under a pan-neuronal synapsin promoter (hsyn-ChR2) in the BA using an injection of adeno-associated virus (AAV) centred on posterior BMA and BLA (*Figure 2—figure supplement 5*). Two weeks later we prepared acute slices of vH from animals performed whole-cell recordings from pyramidal neurons in the axon-rich CA1/ proximal subiculum border (*Figure 2B*). By recording in voltage clamp at –70 mV, we could isolate excitatory currents in response to blue light in ~40% of recorded neurons that were blocked by bath application of the AMPA receptor antagonist NBQX (*Figure 2C*, paired t-test, $t_{(8)}$ = 10.04, p=0.000008, n = 9 neurons). In the same neurons, we could also record inhibitory currents at 0 mV in ~30% of cells that were blocked by the GABA-A receptor antagonist gabazine (*Figure 2E*, paired t-test, $t_{(8)}$ = 11.7, p=1.48 × $10^{-7}$, n = 12 neurons). Thus, BA input makes excitatory and inhibitory connections with vH pyramidal neurons via AMPA and GABA-A receptors.

Our retrograde tracing experiments (*Figure 2A*) suggested that in addition to classic feedforward inhibition (where excitatory axons make connections with local interneurons to disynaptically inhibit pyramidal neurons), BA input also contained axons originating from inhibitory neurons, which would putatively make direct inhibitory connections. To confirm this possibility, we first used a pharmacological approach (*Figure 2—figure supplement 1D–F*). Using mice injected with hsyn-ChR2 in BA as above, we recorded inhibitory currents in vH pyramidal neurons at 0 mV. We first removed feedforward inhibition with bath application of the AMPA receptor antagonist NBQX. Interestingly, while inhibition was completely blocked in a subset of neurons (8/12), in the remaining population inhibitory

currents persisted. This finding suggests that – consistent with our retrograde anatomy – a proportion of this inhibitory input was due to a direct long-range inhibitory projection from the BA. Consistent with this prediction, the remaining current was blocked by bath application of gabazine, indicating that it was a GABA receptor-mediated current.

To test this more explicitly, we again used vGAT-cre mice where cre is expressed only in GABAergic neurons and expressed ChR2 in BA using either a CaMKii promoter – to confine expression to only putative excitatory pyramidal neurons (*Felix-Ortiz et al., 2013*; *Pi et al., 2020*) – or using a cre-dependent cassette to restrict ChR2 only to putative GABAergic neurons (*Seo et al., 2016*). After allowing time for expression, we observed both excitatory and inhibitory axon labelling at the CA1/subiculum border in vH (*Figure 2—figure supplements 3 and 4*), consistent with direct projections from both populations of BA neurons. Consistent with the presence of both excitatory and inhibitory projections, CaMKii+ BA input evoked strong inhibitory currents at 0 mV (*Figure 2G*), but these currents were blocked by bath application of NBQX, showing that the inhibitory currents were a result of solely feedforward inhibition (repeated-measures ANOVA, $F_{(2,4)}$ = 23.4, p=0.006; Tukey's post hoc test, baseline vs. NBQX, $t_{(2)}$ = 4.73, p=0.001, baseline vs. gabazine, $t_{(2)}$ = 4.84, p=0.001, NBQX vs. gabazine, $t_{(2)}$ = 0.12, p=0.90, n = 3 neurons). In contrast, vGAT+ BA input also showed robust input at 0 mV (*Figure 2I*), but this inhibitory current was insensitive to NBQX application, but blocked by gabazine, suggesting a direct inhibitory connection (repeated-measures ANOVA, $F_{(2,10)}$ = 10.03, p=0.004; Tukey's post hoc test, baseline vs. NBQX, $t_{(2)}$ = 0.05, p=0.9, baseline vs. gabazine, $t_{(2)}$ = 4.12, p=0.001, NBQX vs. gabazine, $t_{(2)}$ = 4.16, p=0.001, n = 6 neurons).

Together, these experiments define a novel, direct inhibitory projection from BA to vH. Thus, contrary to previous assumptions, BA provides two parallel projections to pyramidal neurons in vH, one excitatory, and one inhibitory.

## BA excitatory and inhibitory input selectively connects with unique vH output populations

The relatively sparse connectivity in our results above suggests that both excitatory and inhibitory BA input may connect with only a proportion of pyramidal neurons in vH. The CA1/proximal subiculum border of vH is composed of multiple populations of neurons organised as parallel projections (*Figure 3*, *Gergues et al., 2020*; *Naber and Witter, 1998*; *Wee and MacAskill, 2020*). Therefore, we hypothesised that this low connectivity may be an indication that BA input connects differentially with neurons that project to either NAc, PFC or back to BA.

To investigate this possibility, we wanted to directly compare the level of synaptic input from BA onto each projection populations in vH. As the absolute level of input onto a recorded neuron using the CRACM approach is proportional to the number of connected axons times the unitary amplitude of these connections, light-evoked input is dependent on a number of technical variables such as precise location of the injection site, location of recording in vH and the number of ChR2-positive axons. Therefore, it is not possible to compare input onto different populations of neurons across slices and injections (*MacAskill et al., 2014*; *MacAskill et al., 2012*; *Marques et al., 2018*; *Petreanu et al., 2007*). Therefore, we instead compared the relative ChR2-evoked input onto pairs of neighbouring neurons, each of which projected to a different downstream region. Using this approach, we could make a within-experiment comparison of the relative BA input across each of the projection populations, while keeping the stimulus constant (*Petreanu et al., 2007*). In order to carry out this experiment, we again injected ChR2 into BA, but also retrograde tracers into either BA and NAc, or BA and PFC. This allowed us, 2 weeks later, to prepare acute slices and obtain whole-cell recordings from pairs of fluorescently identified neurons in vH projecting to each downstream target. Together, the paired recording of neurons in the same slice and field of view allowed for a comparison of ChR2-evoked synaptic input while controlling for variability in the absolute level of input due to confounds such as injection volume and the exact location in CA1/subiculum.

We first compared excitatory input in voltage clamp at –70 mV as before with pan-neuronal expression of ChR2 using the synapsin promoter. Sequential paired recordings of vH$^{BA}$ and vH$^{NAc}$ neurons showed that light-evoked excitatory BA input was on average equivalent onto both populations (*Figure 4A–C*, Wilcoxon rank-sum, W = 15, p=0.43, n = 9 pairs of neurons). In contrast, paired recordings of vH$^{BA}$ and vH$^{PFC}$ neurons revealed an almost complete lack of excitatory input onto vH$^{PFC}$ neurons (*Figure 4D–F*, Wilcoxon rank-sum, W = 0, p=0.0018, n = 8 pairs of neurons).

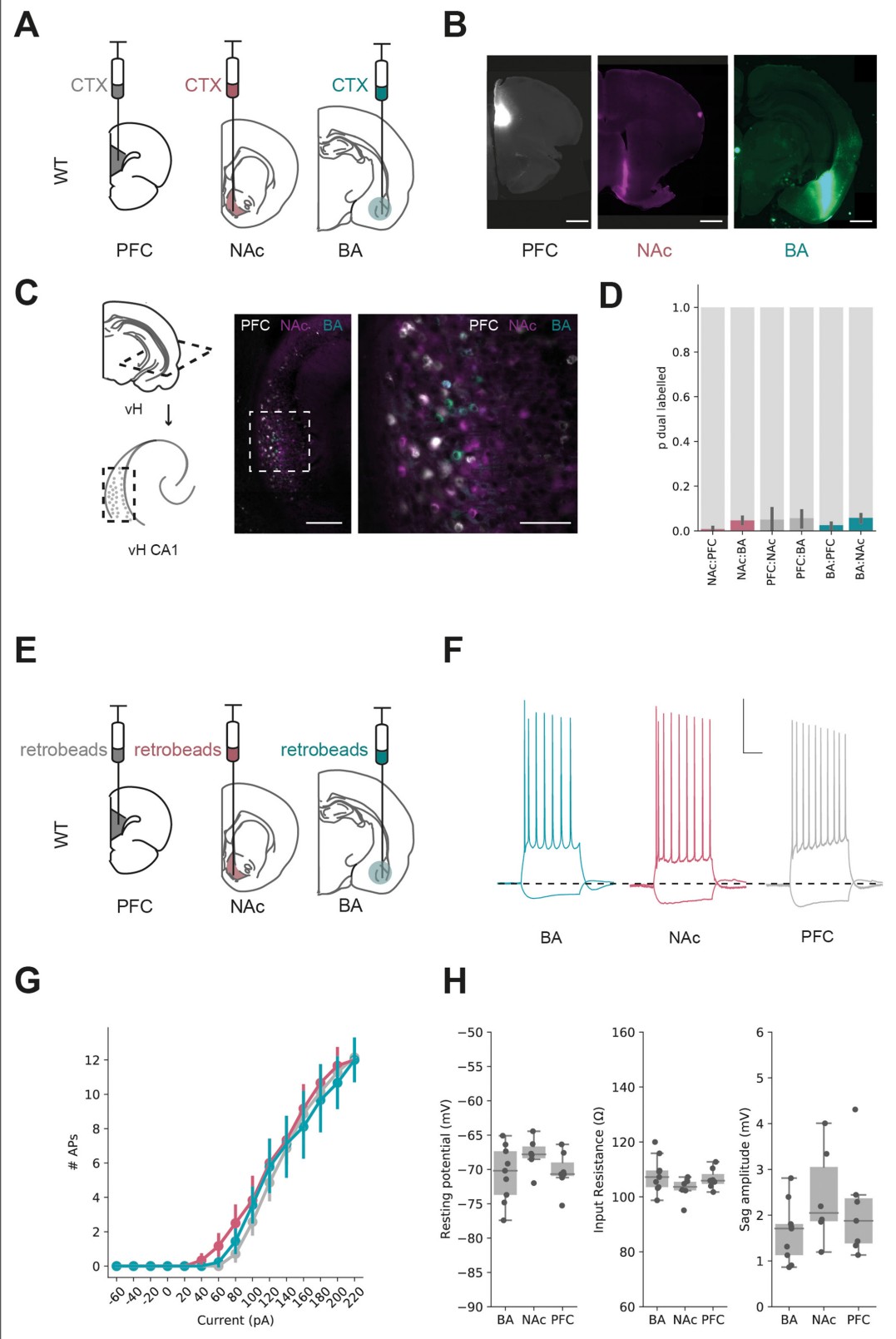

**Figure 3.** Parallel output populations in ventral CA1/subiculum. (**A**) Schematic of experiment, three differently tagged CTXβ tracers were injected into prefrontal cortex (PFC), nucleus accumbens (NAc) and basal amygdala (BA). (**B**) Example injection sites in each region. Scale bar 1 mm. (**C**) Horizontal section of CA1/subiculum in ventral hippocampus (vH) showing interspersed but non-overlapping labelling. Scale bars 300 μm (left), 100 μm (right).

*Figure 3 continued on next page*

*Figure 3 continued*

(**D**) Proportion of neurons labelled with CTXβ injection in NAc (red), BA (green) or PFC (grey) co-labelled with CTXβ from a different region. Note that there is only a small proportion of dual labelled neurons. (**E**) Strategy for electrophysiology recordings – projection populations were fluorescently labelled with retrobead injections into downstream projection areas. (**F**) Examples of positive (+160 pA) and negative (–40 pA) current steps in fluorescently targeted neurons from each population. Scale bar = 30 mV, 100 ms. (**G, H**) No large differences in input/output curve, resting potential, input resistance or sag amplitude across the three populations.

The online version of this article includes the following figure supplement(s) for figure 3:

**Source data 1.** Source data for *Figure 3*.

We next investigated long-range inhibitory input using vGAT-cre mice and expressing cre-dependent ChR2 in BA. Paired recordings of $vH^{BA}$ and $vH^{NAc}$ neurons showed a marked bias of inhibitory input to $vH^{BA}$ neurons, with consistently smaller input onto neighbouring $vH^{NAc}$ neurons (*Figure 4G–I*, Wilcoxon rank-sum, W = 0, p=0.016, n = 7 pairs of neurons). Similarly to excitatory input, pairs of $vH^{BA}$ and $vH^{PFC}$ projecting neurons showed essentially no connectivity from BA to $vH^{PFC}$ neurons (*Figure 4J–L*, Wilcoxon rank-sum, W = 0, p=0.016, n = 7 pairs of neurons).

Overall, these experiments suggest that excitatory input from BA equally targets vH neurons projecting to either NAc or BA, but not with those projecting to PFC. In contrast, inhibitory input from BA preferentially targets vH neurons projecting to BA, has a weak connection to those that project to NAc and again avoids those projecting to PFC. Together, this shows that both excitatory and inhibitory BA input to vH have unique and distinct connectivity patterns with vH output circuitry, and suggests it is well placed to define their differential activity.

## BA excitatory and inhibitory input interacts with local inhibitory circuitry in vH

We next wanted to understand how BA input may interact with the local vH circuit to define activity of the different output populations. vH output populations have been shown to be strongly connected with local interneurons to form both feedforward and feedback inhibitory circuitry, and this connectivity can vary on a cell-type-specific basis (*Lee et al., 2014a*; *Soltesz and Losonczy, 2018*). Thus we next wanted to ask three questions about the layout of the vH circuit and how it is influenced by BA input: (1) Does excitatory and inhibitory BA input connect directly with local interneurons in vH? (2) Do pyramidal neurons from each projection population connect with local interneurons to provide feedback inhibition? (3) Are there differences in how local interneurons connect with pyramidal neurons from different projection populations?

We first asked whether BA excitatory and inhibitory input targeted interneurons in vH. To do this, we combined ChR2 input mapping with an AAV injection in vH to express interneuron-specific fluorescent markers (*Cho et al., 2015*; *Dimidschstein et al., 2016*). This allowed us to record from fluorescently identified interneurons in vH and record light-evoked excitatory or inhibitory input from BA (*Figure 5A–D*). We found similar levels of both excitatory and inhibitory connectivity to input from BA onto local interneurons as we found with pyramidal neurons (in both cases, ~50% of recorded neurons were connected). Thus, both inhibitory and excitatory input from BA connect with local interneurons as well as pyramidal projection neurons in vH.

We next wanted to investigate if $vH^{BA}$ and $vH^{NAc}$ neurons connected to local interneurons to form the basis of a feedback inhibitory circuit (*Lee et al., 2014a*). To do this, we injected a retrogradely transported AAV (AAVretro) in either NAc and BA to express cre recombinase in NAc- or BA-projecting vH neurons, respectively. In the same surgery, we injected a combination of cre-dependent ChR2 and the fluorescent reporter dlx-mRuby into vH. This allowed us to obtain whole-cell recordings from fluorescently identified vH interneurons, while activating neighbouring projection neurons. Voltage-clamp recordings at –70 mV showed robust responses from both $vH^{NAc}$ and $vH^{BA}$ neurons onto local interneurons (~80% of recorded neurons were connected in each condition, *Figure 5E–H*), confirming previous studies suggesting strong feedback inhibition in vH (*Lee et al., 2014a*). For both of these experiments (*Figure 5A–H*), it is important to note that we did not quantitatively compare the level of synaptic input across different conditions due to the limitations of the CRACM approach (see 'Discussion'). However, these experiments confirm that there is robust feedforward and feedback inhibition present in the CA1/subiculum border of vH.

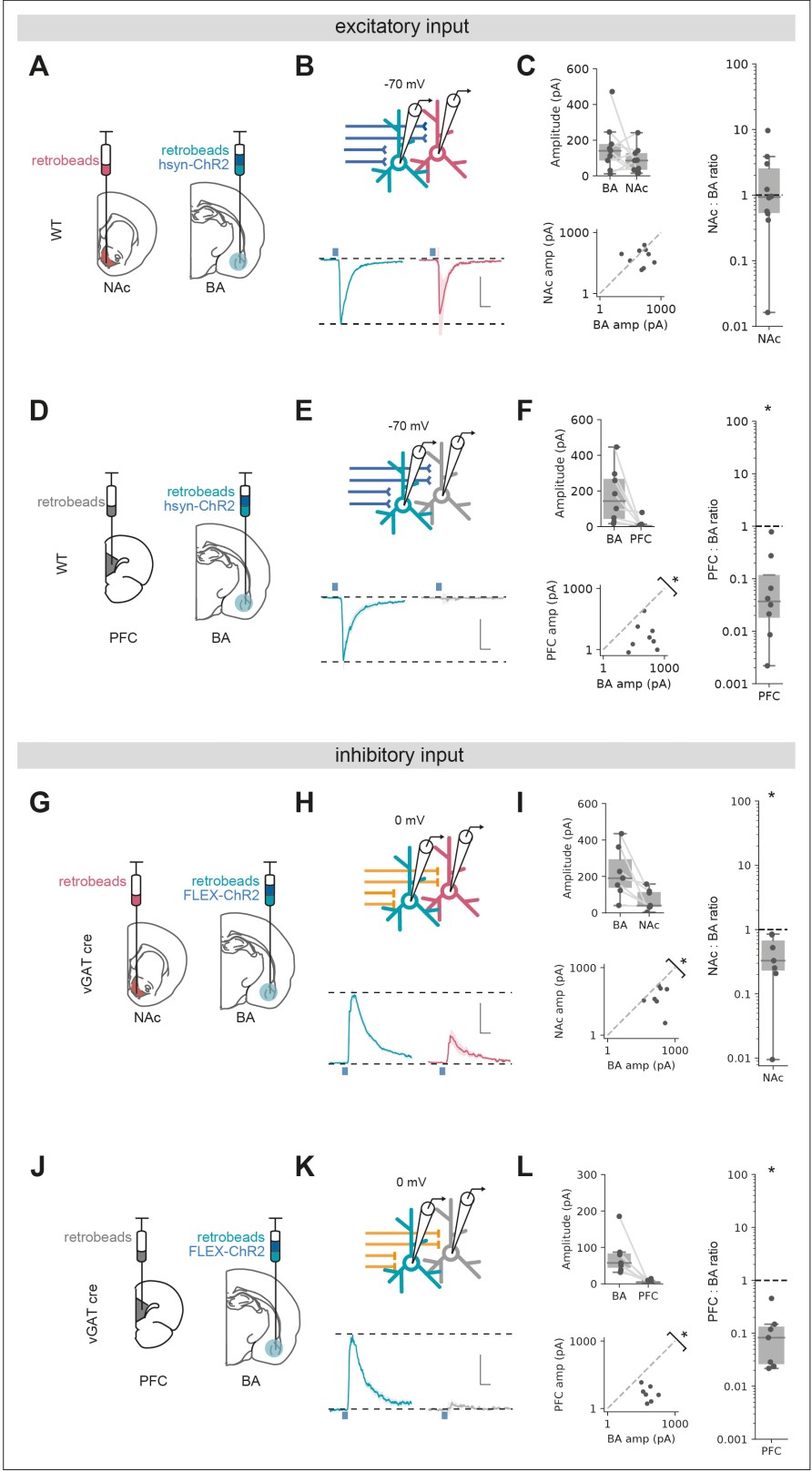

**Figure 4.** Excitatory and inhibitory basal amygdala (BA) input differentially targets ventral hippocampus (vH) output populations. (**A**) Schematic of experiment vH$^{NAc}$ and vH$^{BA}$ neurons was labelled with retrobead injections, and ChR2 was expressed pan-neuronally in BA. (**B**) Paired, fluorescently targeted recordings from neurons in each pathway and recording of light-evoked currents. Top: recording setup. Bottom: average light-evoked currents in

*Figure 4 continued on next page*

*Figure 4 continued*

vH$^{BA}$ (green) and vH$^{NAc}$ (red) neurons. Scale bar = 0.5 vH$^{BA}$ response, 10 ms. (**C**) Summary of amplitude of light-evoked BA input in pairs of vH$^{NAc}$ and vH$^{BA}$ neurons (top). When displayed as a scatter plot (bottom), or as the ratio of vH$^{NAc}$:vH$^{BA}$ (right), the amplitudes cluster on the line of unity, indicating that these populations share equal input. Note log axis. (**D–F**) As (**A–C**) but for pairs of vH$^{BA}$ and vH$^{PFC}$ neurons. Note that when displayed as a scatter and a ratio, both vH$^{PFC}$:vH$^{BA}$ amplitudes are below the line of unity, indicating that input preferentially innervates vH$^{BA}$ neurons. (**G–L**) As (**A–F**) but for inhibitory input from BA isolated by expressing FLEX ChR2 in a vGAT::Cre line. Note that when displayed as a scatter and a ratio, both vH$^{PFC}$ and vH$^{NAc}$ amplitudes are below the line of unity, indicating that inhibitory input preferentially innervates vH$^{BA}$ neurons in both cases. Scale bar = 0.5 vH$^{BA}$ response, 10 ms.

The online version of this article includes the following figure supplement(s) for figure 4:

**Source data 1.** Source data for *Figure 4*.

Finally, we asked if local interneurons differentially innervate vH$^{BA}$ and vH$^{NAc}$ neurons. In order to quantitatively compare input across these two populations, as before we expressed ChR2 in vGAT+ interneurons in vH using a vGAT-cre mouse line and injected different coloured retrobeads into NAc and BA. Two weeks later we then obtained paired, whole-cell recordings from neighbouring vH$^{BA}$ and vH$^{NAc}$ neurons in the same slice and investigated light-evoked inhibitory synaptic input at 0 mV. We found that local inhibitory connectivity was markedly biased towards vH$^{NAc}$ neurons (*Figure 5I–K*), where inhibitory connections onto vH$^{NAc}$ neurons were on average twice the strength of those onto neighbouring vH$^{BA}$ neurons (Wilcoxon rank-sum, W = 2, p=0.006, n = 10 pairs of neurons). Thus, activation of local interneurons in vH, either via direct input from BA or via feedback from local pyramidal neurons, results in biased inhibition of vH$^{NAc}$ neurons and has a much smaller effect of neighbouring vH$^{BA}$ neurons.

This marked asymmetry of local inhibitory connectivity led us to predict that feedforward inhibition activated by excitatory BA input may also differentially impact the two output populations. We tested this using ChR2 expressed in BA under the control of the CaMKii promoter to limit expression to excitatory projections. As before, excitatory input in this experiment was equivalent in neighbouring vH$^{BA}$ and vH$^{NAc}$ neurons (*Figure 5L–N*, Wilcoxon rank-sum, W = 22, p=0.625, n = 10 pairs of neurons). In contrast, and as predicted, feedforward inhibition recorded at 0 mV was markedly biased towards vH$^{NAc}$ neurons (*Figure 5O and P*, Wilcoxon rank-sum, W = 3, p=0.04, n = 10 pairs of neurons).

Together, these experiments show that local interneurons in vH make biased connections onto vH$^{NAc}$ neurons. This biased innervation of interneurons towards vH$^{NAc}$ neurons suggests greater influence of both feedforward inhibition from BA, but also feedback inhibition resulting from activation of local pyramidal neurons.

## A circuit model predicts a role for long-range inhibition in the promotion of vH$^{NAc}$ activity

Our results so far suggest that the connectivity of both excitatory and inhibitory BA input into vH is very specific and interacts with a number of interconnected elements in the local vH circuit. In order to investigate the overall influence of BA input in a more holistic way, we built a simple integrate-and-fire network (*Stimberg et al., 2019*), containing three separate projection populations in vH (to BA, NAc and PFC), local interneurons, excitatory and inhibitory input from BA, and background synaptic input from other structures. We then constrained the connectivity between these groups of neurons using the results of our circuit analysis (*Figure 6A*).

We first looked at excitatory BA input alone and found that this robustly activated vH$^{BA}$ neurons in our model and had no effect on vH$^{PFC}$ activity – consistent with the lack of connectivity to this population (see *Figure 4*). However, there was also a marked lack of vH$^{NAc}$ activity despite these neurons receiving equivalent excitatory synaptic input from BA. This was due to asymmetrical targeting by local inhibition (see *Figure 5*), and thus a combination of feedback and feedforward inhibition effectively silencing vH$^{NAc}$ neurons, despite them receiving excitatory drive.

We next incrementally added increasing proportions of long-range inhibitory input from BA to the model, such that there was co-activation of both long-range inhibitory and excitatory input. We found that increasing inhibitory input resulted in a switch in the activity of the different populations (*Figure 6B and C*). While vH$^{PFC}$ neurons remained silent, vH$^{NAc}$ neuron activity increased as direct

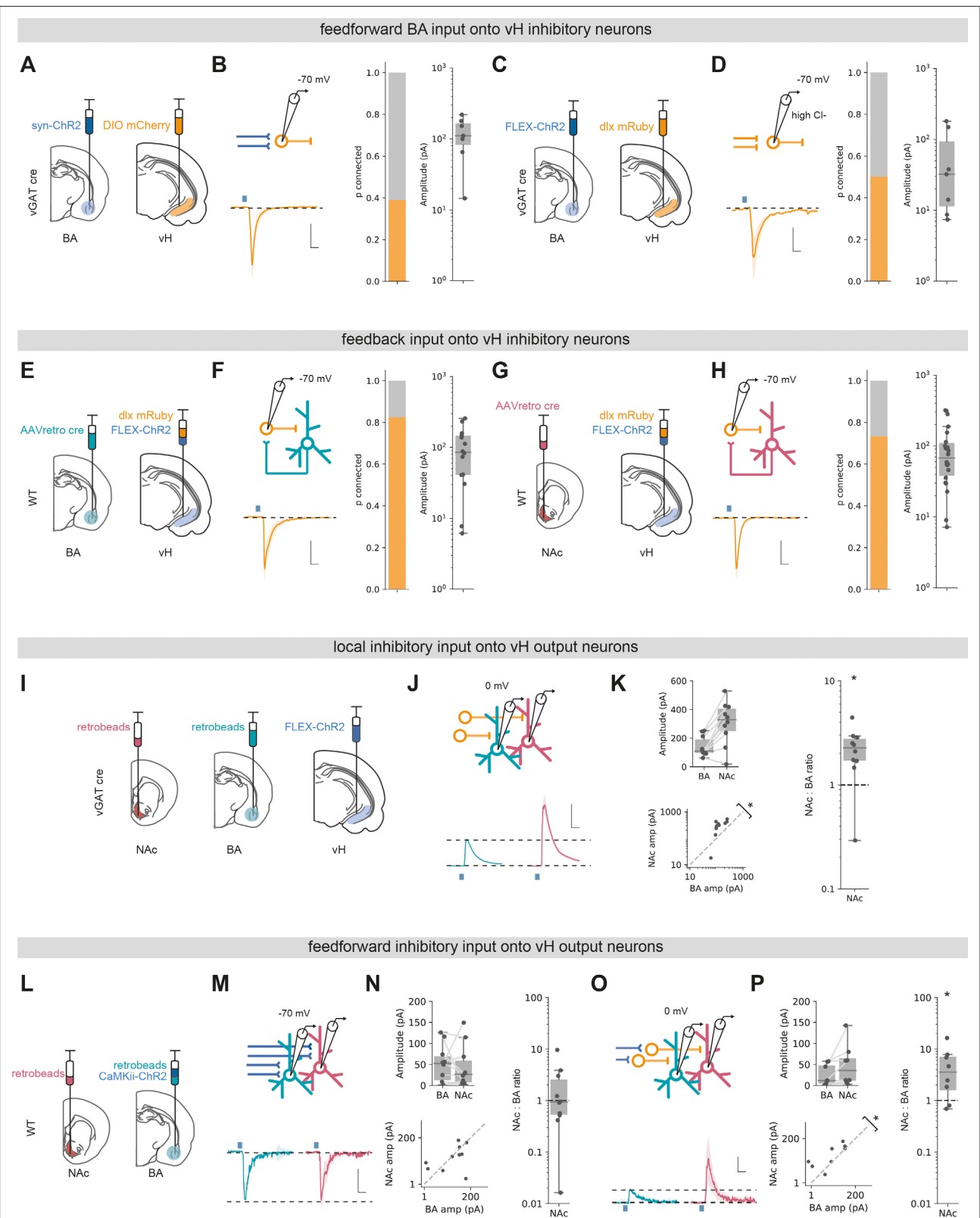

**Figure 5.** Basal amygdala (BA) input interacts with local inhibitory circuitry that is biased towards vH^NAc neurons. (**A**) Schematic of experiment. ChR2 was expressed in BA, and DIO mCherry was expressed in ventral hippocampus (vH) in vGAT:cre mice to label local interneurons. (**B**) Left: recording configuration to record excitatory connectivity at –70 mV (top). Average light-evoked current in interneurons in vH. Scale bar = 50 pA, 10 ms. Right: summary of probability of connection (left) and amplitude of connected currents (right). (**C, D**) As (**A, B**) but for inhibitory input isolated using FLEX ChR2

*Figure 5 continued on next page*

*Figure 5 continued*

expression in vGAT:cre mice as before. Note that recordings were performed in high Cl-, so inward currents were measured at –70 mV. (**E**) Experimental setup for investigating feedback connectivity from vH$^{BA}$ neurons. AAVretro was injected into BA, and FLEX ChR2 and dlx-mRuby into vH to allow recordings from dlx+ interneurons, and measurement of light-evoked currents from vH$^{BA}$ activation. (**F**) Left: recording configuration to record excitatory connectivity at –70 mV (top). Average light-evoked current in dlx+ interneurons in vH. Right: summary of probability of connection (left) and amplitude of connected currents (right). (**G, H**) As (**E, F**) but for feedback input from vH$^{NAc}$ neurons. (**I**) Schematic of experiment, vH$^{NAc}$ and vH$^{BA}$ cells were labelled with injections of retrobeads, while ChR2 was expressed in vH interneurons using FLEX ChR2 in a vGAT::cre mouse. (**J**) Paired, fluorescently targeted recordings from neurons in each pathway at 0 mV and recording of light-evoked currents. Top: recording setup. Bottom: average light-evoked currents in vH$^{BA}$ (green) and vH$^{NAc}$ (red) neurons. Scale bar = 1 vH-BA response, 10 ms. (**K**) Summary of amplitude of light-evoked BA input in pairs of vH$^{NAc}$ and vH$^{BA}$ neurons (top). When displayed as a scatter plot (bottom), or as the ratio of vH$^{NAc}$: vH$^{BA}$ (right), the amplitudes cluster above the line of unity, indicating that local inhibition preferentially innervates vH$^{NAc}$ neurons. Note log axis. (**L–N**) as (**I, J**) but for CaMKii input recorded at –70 mV. Note as in **Figure 3** that there is equal input onto both populations. Scale bar = 0.5 vH$^{BA}$ response, 10 ms. (**O, P**) as in (**M, N**) but recording at 0 mV to isolate feedforward inhibition. Note that the amplitudes cluster above the line of unity, indicating that feedforward inhibition preferentially innervates vH$^{NAc}$ neurons. Scale bar = 1 vH-BA response, 10 ms.

The online version of this article includes the following figure supplement(s) for figure 5:

**Source data 1.** Source data for *Figure 5*.

inhibition increased, and vH$^{BA}$ neuron activity decreased. This difference peaked around 40% long-range inhibition, where vH$^{BA}$ neurons were effectively silent, and vH$^{NAc}$ neurons were firing robustly. This was due to long-range inhibition efficiently removing feedforward and feedback inhibition onto vH$^{NAc}$ neurons (**Figure 6D**) – both by direct inhibition of local interneuron activity, but also by inhibiting vH$^{BA}$ neurons that provide the bulk of feedback inhibitory drive. This effect was robust across a wide range of feedforward and feedback connectivity (**Figure 6—figure supplement 1**), was robust to large proportions of overlap between the different projection populations (**Figure 6—figure supplement 2**) and was independent of the postsynaptic mechanism underlying the differences in overall input – either postsynaptic amplitude or connection probability (**Figure 6—figure supplement 3**; and see 'Discussion').

This circuit analysis suggests that specific connectivity of excitatory BA input into vH may not be the major determinant of vH$^{BA}$ and vH$^{NAc}$ neuron activity. In fact, it is the presence of direct inhibitory input from BA that defines which projection population is active. With no inhibition present, activity

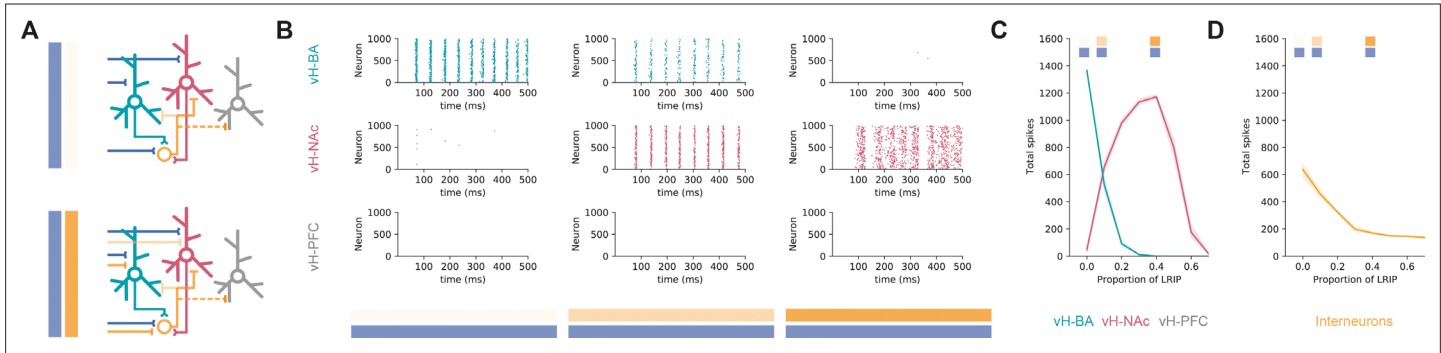

**Figure 6.** Co-activation of inhibitory and excitatory input switches ventral hippocampus (vH) activity from vH$^{BA}$ to vH$^{NAc}$. (**A**) Schematic of integrate-and-fire model. Three populations of projection neurons (vH$^{NAc}$, red; vH$^{BA}$, green; vH$^{PFC}$, grey) and local interneurons (orange) are innervated by excitatory (blue, top) as well as inhibitory (orange, bottom) basal amygdala (BA) input. Connectivity is defined from results in previous figures. (**B**) Increasing the proportion of inhibitory relative to excitatory BA input has opposite effects on vH$^{BA}$ and vH$^{NAc}$ spiking. Each graph shows a raster of spiking for each neuron across a 500 ms period. Note high vH$^{BA}$ spiking with no inhibitory input, and high vH$^{NAc}$ spiking with high inhibitory input. vH$^{PFC}$ neurons never fire as they are not innervated by BA and only receive background input. (**C**) Summary of pyramidal neuron activity. With increasing inhibitory input, activity shifted from vH$^{BA}$ to vH$^{NAc}$ neurons. Markers indicate proportions plotted in (**B**). (**D**) Long-range inhibition reduces local interneuron firing, removing preferential feedback inhibition onto vH$^{NAc}$ neurons, allowing them to fire.

The online version of this article includes the following figure supplement(s) for figure 6:

**Source data 1.** Source data for *Figure 6*.

**Figure supplement 1.** The switch in vH$^{BA}$ and vH$^{NAc}$ activity is robust over a wide range of feedforward and feedback connectivity.

**Figure supplement 2.** The switch in vH$^{BA}$ and vH$^{NAc}$ activity is robust to collateralisation of output projections.

**Figure supplement 3.** The switch in vH$^{BA}$ and vH$^{NAc}$ activity is robust to postsynaptic specialisation.

is confined to a reciprocal projection back to BA; however, when inhibition is present there is a switch to increased activity to NAc.

## BA input to vH can support RTPP via activation of vH[NAc] neurons

A hallmark of activation of vH[NAc] activation is the ability to promote real-time place preference (RTPP; *Britt et al., 2012*; *LeGates et al., 2018*). The results of our circuit modelling suggested that co-activation of BA inhibitory and excitatory input to vH results in vH[NAc] activation. We reasoned that BA input to vH may also support RTPP via activation of vH[NAc] neurons in vivo, and that this would depend on the co-activation of inhibitory as well as excitatory BA projections.

We tested if activation of both excitatory and inhibitory BA input supported RTPP by unilaterally injecting either GFP, or ChR2 under the pan-neuronal synapsin promoter into BA and implanting optical fibres in vH (*Figure 7A*). We then carried out an RTPP test where one side of a rectangular arena was paired with 20 Hz blue light stimulation of BA terminals in vH. Consistent with our circuit analysis showing BA input activating vH[NAc] neurons, this stimulus supported RTPP in ChR2-expressing animals compared to GFP controls (t-test, $t_{(5.9)}$ = 2.61, p=0.041, GFP n = 6 mice, ChR2 n = 8 mice), with no change in the total distance moved during the session (*Figure 7B, C* and t-test, $t_{(9.2)}$ = 1.27, p=0.23).

From our circuit model, we predicted that this RTPP should be abolished by a reduction in the activity of vH[NAc] neurons. We next directly tested this using a combination of optogenetic RTPP to activate BA input, and the *Kappa Opioid Receptor Designer* receptor exclusively activated by designer drugs (KORD) to reversibly inhibit vH[NAc] neurons (*Vardy et al., 2015*). We first tested the efficacy of KORDs expressed in vH[NAc] neurons and confirmed that the KORD agonist salvinorin B (SalB) hyperpolarised vH[NAc] neurons and resulted in a decrease in current-induced action potential firing (*Figure 7D–F*, *Figure 7—figure supplement 1*). We next combined this KORD-mediated inhibition with the optogenetic RTPP assay. We expressed pan-neuronal ChR2 in BA, KORDs in vH[NAc] neurons, and implanted an optical fibre unilaterally in vH (*Figure 7G*). We then carried out the RTPP assay 15 min after a subcutaneous injection of either SalB or vehicle control (DMSO, *Figure 7H–K*). We found that after DMSO injection there was still robust RTPP in both control and KORD-expressing mice. After SalB, control animals again still had robust RTPP. However, after injection of SalB in KORD-expressing animals, RTPP was abolished (mixed-effect ANOVA, effect of group [control vs. KORD]: $F_{(1,14)}$ = 15.97, p=0.001, effect of drug [SalB vs. DMSO]: $F_{(1,14)}$ = 15.06, p=0.002, interaction: $F_{(1,14)}$ = 7.45, p=0.016; post hoc paired t-test: control DSMO vs. SalB, $t_{(8)}$ = 1.1, p=0.3, n = 9 mice, KORD DMSO vs. SalB, $t_{(6)}$ = 4.62, p=0.004, n = 7 mice). Together, these experiments support our circuit model, where co-activation of both excitatory and inhibitory BA input to vH supports RTPP through the activation of vH[NAc] neurons.

## Excitatory BA input to vH supports RTPP only when vH[BA] activity is inhibited

In contrast to activation of both excitatory and inhibitory BA input into vH, another prediction from our circuit modelling is that excitatory BA input alone would not activate vH[NAc] neurons, and thus would not support RTPP. We tested this prediction using ChR2 expressed under the CaMKii promoter to target only excitatory BA input to vH (see *Figure 2*). We injected either GFP or ChR2 under the CaMKii promoter in BA and implanted an optical fibre in vH before carrying out an RTPP assay as before (*Figure 8A*). Consistent with the predictions from our circuit analysis, this assay showed that the light stimulus was unable to support RTPP in either GFP- or ChR2-expressing animals (*Figure 8B and C* and t-test, $t_{(6.4)}$ = 0.40, p=0.70, GFP n = 4 mice, ChR2 n = 7 mice) and was again accompanied by no change in distance travelled (t-test, $t_{(6.9)}$ = 0.08, p=0.94).

Our reasoning for this lack of RTPP was that excitatory BA input results in vH[BA] neuron activity, and this recruits strong local feedback inhibition that preferentially reduces the activity of vH[NAc] neurons (*Figure 6*) that are required to support RTPP (*Figure 7*). We therefore hypothesised that reducing vH[BA] neuron activity (in effect mimicking the effect of the direct BA inhibitory projection) may increase vH[NAc] activity and support RTPP from only excitatory BA input. This reasoning was supported by our circuit model, where removing vH[BA] activity increased the activity of vH[NAc] neurons when no BA inhibitory input was present (*Figure 8—figure supplement 1*).

To test this hypothesis, we first ensured that KORD-expressing vH[BA] neuron excitability was inhibited by bath application of SalB (*Figure 8D–F*, *Figure 7—figure supplement 1*). Next, we injected

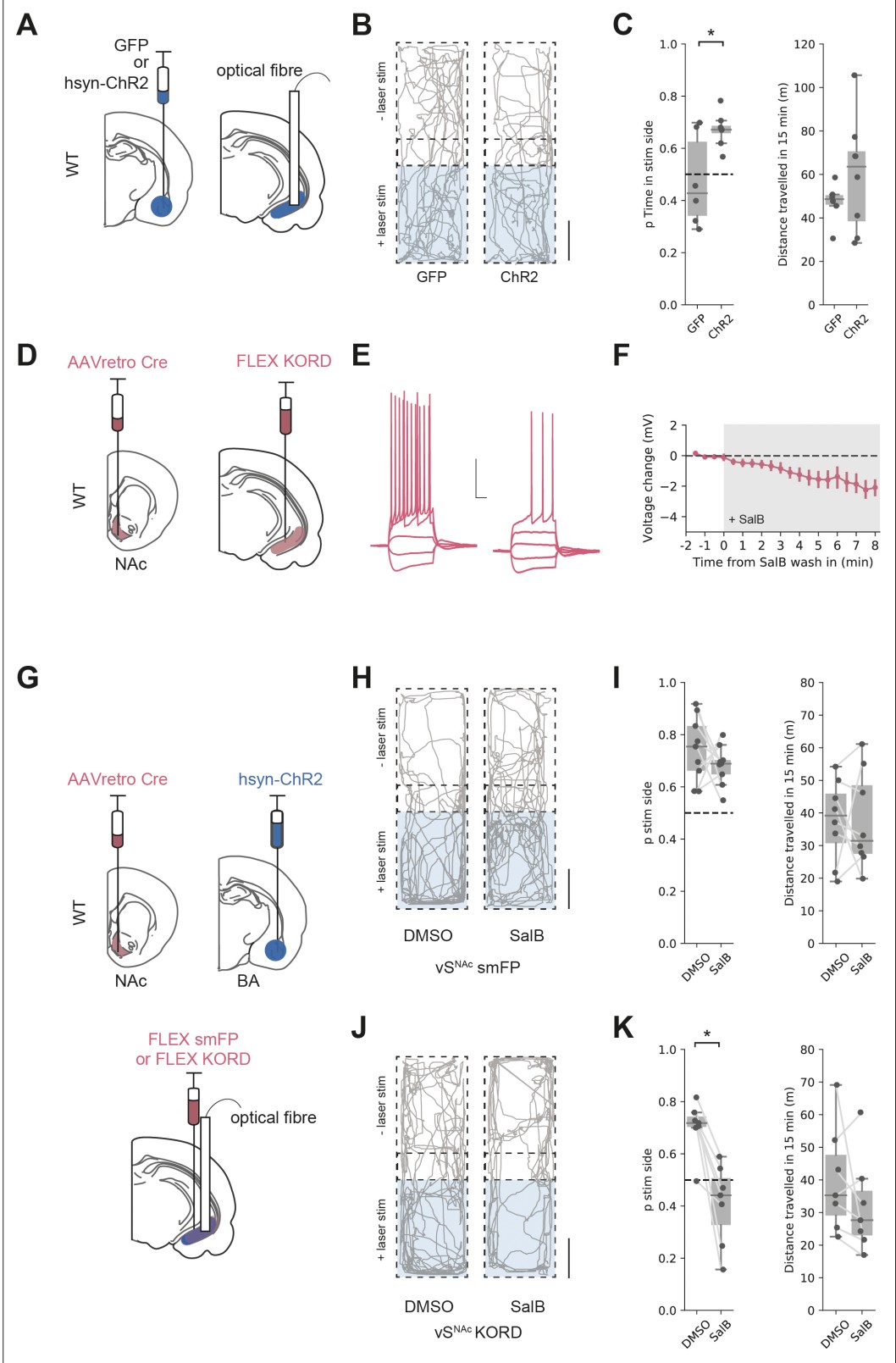

**Figure 7.** Basal amygdala (BA) input supports real-time place preference (RTPP) dependent on vH^NAc neurons. (**A**) Schematic of experiment. GFP or pan-neuronal ChR2 were expressed in BA and an optic fibre implanted in ventral hippocampus (vH). (**B**) RTPP assay. One side of a chamber was paired with 20 Hz blue light stimulation. Example trajectories of GFP (left) and ChR2 (right)-expressing animals over the 15 min RTPP session. Note increased

*Figure 7 continued*

occupancy of light-paired (stim) side in ChR2 animals. Scale bar = 15 cm. (**C**) Summary of RTPP. Left: proportion of time spent on stim side (left) and total distance travelled (right) in GFP and ChR2 animals. Note consistent preference for stim side in ChR2 animals. (**D**) Strategy to express KORD in vH$^{NAc}$ neurons. (**E, F**) Bath application of salvinorin B (SalB) (100 nm) hyperpolarises KORD-expressing vH$^{NAc}$ neurons and reduces AP firing. See *Figure 7— figure supplement 1* for full quantification. Scale bar = 30 mV, 100 ms. (**G**) Schematic of strategy to inhibit vH$^{NAc}$ neurons during BA input-driven RTPP. (**H, I**) As (**B, C**) but comparing the effect of either DMSO (vehicle) or SalB (KORD agonist) injections 15 min before testing in control mice. Note consistent RTPP in both conditions indicating no effect of SalB in control mice. (**J, K**) As (**H, I**), but in mice expressing KORD in vH$^{NAc}$ neurons. Note loss of RTPP in SalB-injected mice compared to controls.

The online version of this article includes the following figure supplement(s) for figure 7:

**Source data 1.** Source data for *Figure 7*.

**Figure supplement 1.** Salvinorin B (SalB) wash in reduces activity of KORD-expressing neurons.

**Figure supplement 2.** Example injection sites for KORD experiments.

**Figure supplement 3.** Histology for behavioural experiments in *Figure 7*.

ChR2 under the CaMKii promoter in BA to target only excitatory input into vH. In the same surgery, we combined this with an injection of AAVretro cre in BA and cre-dependent KORD in vH to target vH$^{BA}$ neurons and implanted an optical fibre unilaterally in vH (*Figure 8J*). After allowing for expression, we then performed the RTPP assay 15 min after injection of either SalB or vehicle control as before (*Figure 8H–K*). Consistent with our previous results, there was no RTPP in either group after DMSO injections or in control animals after SalB injection. However, after SalB injections in KORD-expressing animals, light stimulation now supported RTPP (mixed-effect ANOVA, effect of group [control vs. KORD]: $F_{(1,14)}$ = 3.56, p=0.08, effect of drug [SalB vs. DMSO]: $F_{(1,14)}$ = 3.0, p=0.11, interaction: $F_{(1,14)}$ = 10.85, p=0.005; post hoc paired t-test: control DSMO vs. SalB, $t_{(8)}$ = 1.01, p=0.34, n = 9 mice, KORD DMSO vs. SalB, $t_{(6)}$ = 3.14, p=0.02, n = 7 mice). This was accompanied by a subtle but significant decrease in the distance travelled, reflecting mice increasing quiet resting and grooming bouts in the preferred chamber (paired t-test: control DSMO vs. SalB, $t_{(8)}$ = 0.66, p=0.52, n = 9 mice, KORD DMSO vs. SalB, $t_{(6)}$ = 4.15, p=0.01, n = 7 mice).

This experiment supports our hypothesis that vH$^{NAc}$ activity and hence RTPP is crucially dependent on the activity of both excitatory and inhibitory input from BA. Excitatory BA input to vH can only support RTPP if accompanied by inhibition of BA-projecting vH neurons, in effect mimicking the effect of BA inhibitory input on the circuit. Our model predicts that this reduction in vH$^{BA}$ activity removes local feedback inhibition (*Figure 8—figure supplement 1*) and allows excitatory BA input to drive vH$^{NAc}$ activity, which can support place preference.

## Discussion

In this study, we have defined a novel long-range inhibitory projection from BA to vH. We show that this novel projection exists in concert with a parallel excitatory projection, and that the presence of its inhibitory influence can dramatically shift vH output in response to BA activity. While excitation alone preferentially drives a reciprocal projection back to BA, co-activation of both excitatory and inhibitory input preferentially drives a separate projection to NAc, which can support place-value associations.

We found that in addition to classically described excitatory input from BA to vH, there was also direct inhibitory projection (*Figures 1 and 2*). Excitatory input from BA to vH has been widely studied and is distributed across a large range of subnuclei, ranging from the MEA to the BLA, and well as cortical amygdala (*McDonald and Mott, 2017*). Each of the distinct nuclei of the amygdala are thought to control various aspects of cue-dependent learning and carry out unique roles during behaviour. Increasingly, function has been assigned to BA based on anatomical location. For example, anterior basolateral, basomedial and central amygdala have unique contributions to fear learning and extinction (*Adhikari et al., 2015*; *Ciocchi et al., 2010*; *Kim et al., 2016*; *LeDoux, 2000*), while more posterior and medial regions of BA are increasingly associated with reward learning, value calculations and prosocial behaviours (*Chen et al., 2019*; *Kim et al., 2016*; *Lutas et al., 2019*; *Malvaez et al., 2019*; *Pi et al., 2020*; *Shemesh et al., 2016*). However, the role within each of these nuclei is also diverse – with interspersed neurons involved in encoding behaviour across a wide range of different

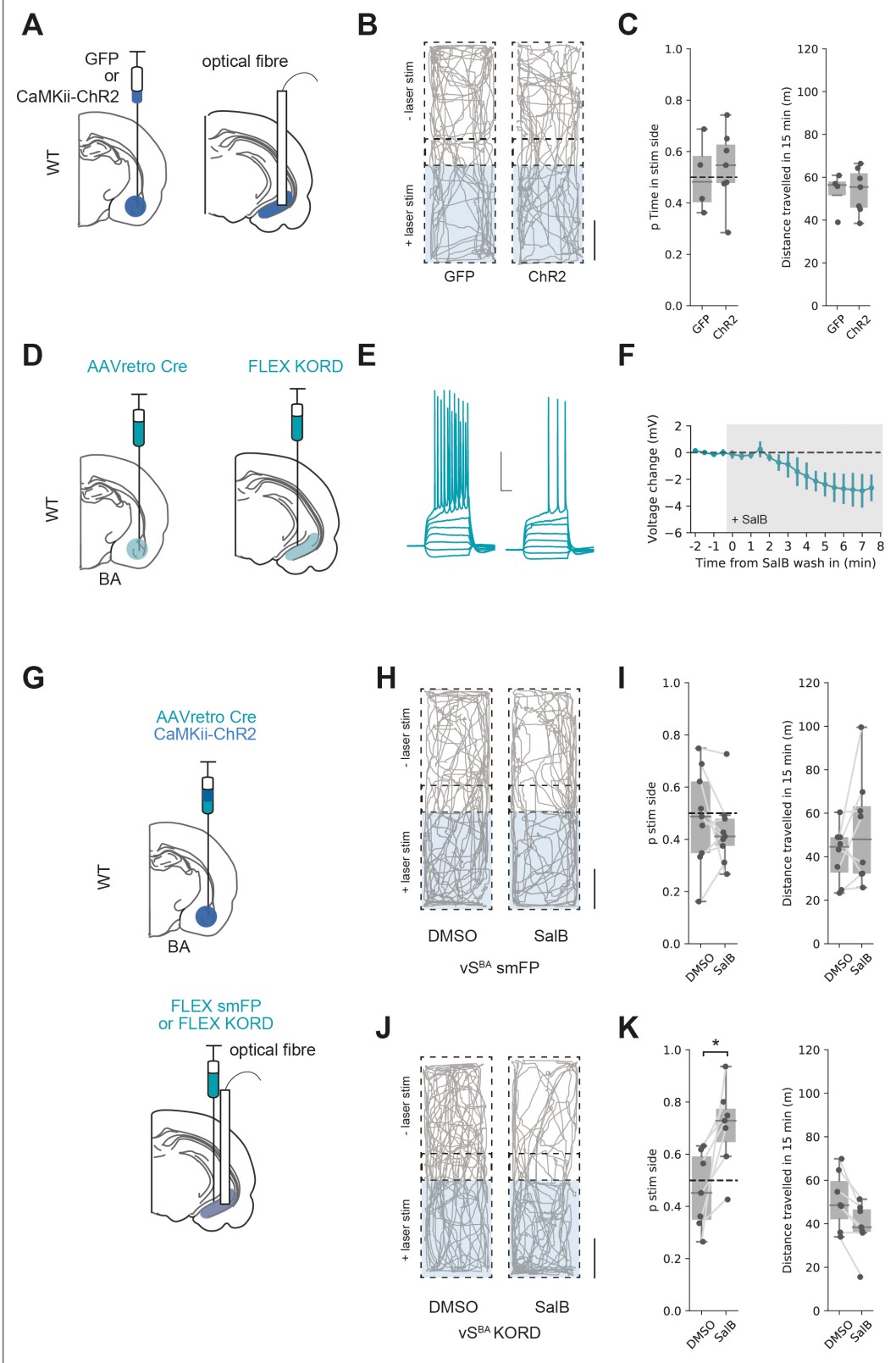

**Figure 8.** Excitatory basal amygdala (BA) input supports real-time place preference (RTPP) only after inhibition of vH$^{BA}$ neurons. (**A**) Schematic of experiment. GFP or excitation-specific CaMKii ChR2 were expressed in BA and an optic fibre implanted in ventral hippocampus (vH). (**B**) RTPP assay. One side of a chamber was paired with 20 Hz blue light stimulation. Example trajectories of GFP (left) and ChR2 (right)-expressing animals over the 15 min RTPP

*Figure 8 continued on next page*

*Figure 8 continued*

session. Note lack of preference for light-paired (stim) side in either group. Scale bar = 15 cm. (**C**) Summary of RTPP. Left: proportion of time spent on stim side (left) and total distance travelled (right) in GFP and ChR2 animals. Note lack of preference for stim side in either condition. (**D**) Strategy to express KORD in vH[BA] neurons. (**E, F**) Bath application of salvinorin B (SalB) (100 nm) hyperpolarises KORD-expressing vH[BA] neurons and reduces AP firing. See *Figure 6—figure supplement 1* for full quantification. Scale bar = 30 mV, 100 ms. (**G**) Schematic of strategy to inhibit vH[BA] neurons during BA input-driven RTPP. (**H, I**) As (**B, C**) but comparing the effect of either DMSO (vehicle) or SalB (KORD agonist) injections 15 min before testing in control mice. Note lack of RTPP in both conditions indicating no effect of SalB in control mice. (**J, K**) As (**H, I**), but in mice expressing KORD in vH[BA] neurons. Note induction of RTPP in SalB-injected mice compared to controls.

The online version of this article includes the following figure supplement(s) for figure 8:

**Source data 1.** Source data for *Figure 8*.

**Figure supplement 1.** Removing vH[BA] activity from the integrate of fire model increases vH[NAc] activity in response to excitatory but not excitatory and inhibitory basal amygdala (BA) input.

**Figure supplement 2.** Histology for behavioural experiments in *Figure 8*.

situations (*Beyeler et al., 2016*; *Felix-Ortiz et al., 2013*; *Felix-Ortiz and Tye, 2014*; *Gründemann et al., 2019*; *Kim et al., 2016*; *Namburi et al., 2015b*). We found that the BA inhibitory projection arose from GABAergic neurons interspersed between excitatory projection neurons throughout the entire extent of the BA (*Figure 2—figure supplement 1*). Thus, it will be important to systematically investigate the synaptic targeting and behavioural contribution of the input from different nuclei separately. However, in addition it will also be important to assess the differential contribution of excitatory and inhibitory drive, most likely through the use of intersectional genetic and anatomical approaches (*Fenno et al., 2014*; *Kim et al., 2016*).

Our results suggest that inhibitory input from BA to vH may be important for motivated behaviour, in particular we show that co-activation of both excitatory and inhibitory projections from BA, and not excitation alone, is essential for promoting place preference (*Figures 7 and 8*). Long-range inhibitory projections from classical excitatory projection areas have been increasingly identified as having a key role in shaping circuit output and for defining motivated behaviour. For example, functional inhibitory projections from PFC to NAc (*Lee et al., 2014b*), and BA to PFC (*Seo et al., 2016*) have both been shown to modulate value-based and reward behaviour, including the support of RTPP and aversion. The hippocampus also receives long-range inhibitory input from numerous regions including entorhinal (*Basu et al., 2016*; *Melzer et al., 2012*), septum (*Schlesiger et al., 2021*) and PFC (*Malik et al., 2021*). While these studies focussed on dorsal hippocampal circuitry and a role for these projections in memory and navigation, due to the known dichotomy between dorsal and ventral hippocampal function (*Fanselow and Dong, 2010*; *Strange et al., 2014*), it would be interesting to investigate the presence and function of such long-range inhibitory projections into vH. In particular, whether a role in motivated behaviour and place preference was specific to BA input or due to the dorsoventral location of this input in hippocampus. Interestingly, long-range inhibition from entorhinal cortex, septum and PFC all preferentially target interneurons and avoid pyramidal neurons (*Basu et al., 2016*; *Melzer et al., 2012*; *Schlesiger et al., 2021*). In contrast, our data show that BA long-range inhibition connects with both interneurons and pyramidal neurons (*Figure 4*), similar to that seen in long-range inhibitory projections from BA to PFC (*Seo et al., 2016*). This suggests that there may at least in part be interesting input-specific connectivity across the different long-range inhibitory inputs into hippocampus.

We investigated the synaptic and circuit basis by which BA input could promote such motivated behaviour. The vH is increasingly viewed as being composed as a series of parallel output streams, where pyramidal neurons in the CA1/subiculum border are composed of multiple populations each projecting to a distinct downstream region including the NAc, the PFC and the BA. Each projection population in vH underlies a unique role during behaviour. In particular, vH[NAc] neurons have been shown to be key for motivated behaviour, and the association of reward with a particular place or time (*Britt et al., 2012*; *Ciocchi et al., 2015*; *LeGates et al., 2018*; *Okuyama et al., 2016*; *Trouche et al., 2019*). We found that both excitatory and inhibitory input from BA made specific connections onto each of these projection populations (*Figures 4–6*), such that the balance of excitation and inhibition from BA into vH is well placed to determine their relative activity. Excitatory input alone preferentially

activated vH[BA] neurons, while excitatory and inhibitory input together preferentially activated vH[NAc] neurons (*Figure 6*). Thus, BA input is well placed to define the activity of specific vH output pathways in response to a particular environment, state or task. More specifically, the level of inhibitory input form BA can control RTPP by defining the activity of vH[NAc] neurons (*Figures 7 and 8*).

It is interesting to note, however, that there is overlap between different projection populations in vH. While roughly 80–90 % of neurons are thought to project to a single downstream region (*Figure 3*, *Gergues et al., 2020*; *Naber and Witter, 1998*; *Wee and MacAskill, 2020*), a proportion of vH neurons collateralise and project to two or more regions. In this study, we recorded from only single-labelled neurons after injection into two downstream regions (*Figures 3–5*), but as the efficiency of retrograde labelling is not 100% we cannot discount the fact that neurons in our dataset may project to more than one region not labelled by our injections. While the large differences in synaptic connectivity across projection populations (*Figures 4 and 5*) reinforce the idea of parallel projection populations in vH, due to their scarcity we did not explicitly address the connectivity of this small population of collateralising neurons. This is therefore an interesting future direction. Importantly, however, using our circuit model we found that the switch in activity from vH[BA] to vH[NAc] populations was robust despite large overlap of each projection population (*Figure 6—figure supplement 2*).

When considering the possibility of collateralising vH neurons, it is important to consider the distribution of projection neurons along the transverse (near CA2, to near subiculum) axis. Dual-projecting neurons are much more prominent in the proximal CA1 (at the CA1/CA2 border; *Naber and Witter, 1998*; *Wee and MacAskill, 2020*). This part of CA1 is preferentially associated with place coding and spatial navigation (*Ciocchi et al., 2015*; *Henriksen et al., 2010*). We focussed our investigations in distal CA1 (at the CA1/subiculum border) as this is where we found the most consistent excitatory and inhibitory input from BA (*Figure 2—figure supplements 3 and 4*; *McDonald and Mott, 2017*), and this is where the majority of long-range projection neurons are found (*Figure 3*; *Naber and Witter, 1998*; *Wee and MacAskill, 2020*). In this part of the hippocampus, dual-projection neurons are rarer (*Naber and Witter, 1998*), and this difference in cellular properties coincides with a preferential role of distal CA1 in non-spatial and object-place associations (*Igarashi et al., 2014*; *Nakamura et al., 2013*). Therefore, in the future it will be important to explicitly investigate the properties of these dual-projection neurons and also how their function and connectivity change along the transverse axis.

In addition to the role of BA and vH in value-based and motivated behaviour, multiple studies have examined the role of excitatory BLA input into vH in the generation of anxiety-like behaviour (*Felix-Ortiz et al., 2013*; *Pi et al., 2020*). The vH has a key role in the generation of appropriate behaviour in anxiogenic environments (*Gray and McNaughton, 2003*; *Kjelstrup et al., 2002*; *McHugh et al., 2004*). This is thought to be achieved both by resolving approach-avoidance conflict during decision-making via vH[PFC] projection neurons (*Padilla-Coreano et al., 2016*; *Sanchez-Bellot and MacAskill, 2021*), but recently also via generation of a specific anxiogenic state defined via projections to the lateral hypothalamus (LH; *Jimenez et al., 2018*). In our study, we detected only minimal excitatory or inhibitory BA input onto vH[PFC] neurons (*Figure 4*), suggesting that innervation from other local or long-range afferent regions may be key for this behavioural role (*Sanchez-Bellot and MacAskill, 2021*). However, BA input does innervate vH[LH] neurons (*Gergues et al., 2020*; *Wee and MacAskill, 2020*), and thus it is interesting to note the possibility that the anxiogenic influence of excitatory, anterior BLA input (*Felix-Ortiz et al., 2013*; *Pi et al., 2020*) may be via this distinct circuit. vH[LH] neurons are present in more distal areas of ventral subiculum, with only a minority present in the CA1/ proximal subiculum border region considered in this study (*Wee and MacAskill, 2020*). However, how BA input interacts with distal subicular circuits that project to distinct downstream regions including hypothalamus and retrosplenial cortex (*Cembrowski et al., 2018*; *Kim and Spruston, 2012*), and how inhibitory and excitatory input interact with this circuit is an interesting future direction.

Our study focussed on the postsynaptic influence of BA inhibitory projections, and the cellular properties of these projection neurons remain unknown. Anatomical studies have suggested that BA inhibitory projections are preferentially observed in somatostatin (SOM)- and neuropeptide Y (NPY)-expressing neurons (*McDonald et al., 2012*; *McDonald and Zaric, 2015*), and almost completely absent in parvalbumin (PV)- and vasoactive intestinal peptide (VIP)-expressing neurons. Consistent with this, we found a proportion of SOM-positive neurons in BA that project to vH (*Figure 2—figure supplement 2*). Thus, there is the potential for inhibitory input to be from both specific nuclei in BA

(*Figure 1*, *Figure 2—figure supplement 1*), but also different genetically defined populations of inhibitory neurons, as is seen for excitatory amygdala projections (*Kim et al., 2016*). Similarly, in our study we did not differentiate BA input onto different types of inhibitory interneuron in vH. There is enormous diversity of interneuron types throughout the hippocampus (*Group et al., 2008*), each of which is involved in distinct parts of the circuit calculation – such as dendritic-targeting SOM- and VIP-expressing neurons, perisomatic PV-expressing interneurons and cholecystokinin (CCK)-expressing interneurons. Inhibitory input from entorhinal cortex preferentially innervates CCK interneurons (*Basu et al., 2016*), while input from PFC specifically innervates VIP interneurons (*Malik et al., 2021*). Thus, how BA input differentially innervates these populations is an important and interesting future question.

Finally, it is important to note that we investigated the connectivity of this circuit at a steady state, and all of our slice physiology was performed in animals that had only experienced their home cage environment. Therefore, it is unknown how this circuit may be updated by experience and new learning, and the plasticity mechanisms that might underlie this updating. The reciprocal connection from vH to BA has been shown to undergo robust plasticity (*Bazelot et al., 2015*), and BA circuitry rapidly updates in response to learning cue associations (*Beyeler et al., 2016*; *Namburi et al., 2015b*; *Namburi et al., 2015a*). Therefore, an important future direction will be to understand how the BA-vH circuit is altered by learning and novel experience, and how this plasticity influences the relative targeting of excitatory and inhibitory connections onto each of the vH projection populations.

## Technical limitations of CRACM

In this study, we used CRACM to investigate the connectivity between BA and vH. We utilised this technique as axons from BA are severed during the slicing process which renders them unable to be electrically stimulated. In addition, the CRACM technique allowed us to restrict our analysis to specific genetically defined excitatory or inhibitory input. However, there are multiple caveats associated with the CRACM technique that must be taken into account when interpreting such experiments. First, in the standard CRACM setup, light-evoked currents in postsynaptic neurons are heavily dependent on the number of connections with ChR2-positive axons, as well as the amplitude of the postsynaptic response at each connected synapse (*MacAskill et al., 2014*; *MacAskill et al., 2012*; *Marques et al., 2018*; *Petreanu et al., 2007*). Thus, the absolute size of a ChR2 response is crucially dependent on the number of infected axons and the level of ChR2 expression in each axon. This makes a comparison across experiments extremely challenging. To mitigate this, in our study we compared the light-evoked response across two neighbouring neurons in the same slice, one projecting to each downstream region under investigation. By comparing responses to the same stimulus in each neighbouring neuron, we could quantitatively compare the relative input onto each cell type across experiments. Importantly where these paired recordings were not possible – such as when investigating interneuron connectivity in *Figure 5A–H* and a quantitative comparison was not possible, we could only infer qualitative connectivity. In this case, we used a circuit model to investigate the consequences of systematically altering this connectivity (*Figure 6—figure supplement 1*) and found that the behaviour of the circuit was consistent across a broad range of connectivity. A second related issue is that the basic CRACM technique cannot differentiate the postsynaptic mechanism underlying differences in input across cell types. For example, in *Figure 4* we could not differentiate if the greater input onto vH$^{BA}$ neurons compared to neighbouring vH$^{PFC}$ neurons is due to differences in connection probability or differences in the strength of these connections. Using our circuit model we showed that for the simple circuit layout we consider for this study the precise postsynaptic mechanism did not influence the circuit properties (*Figure 6—figure supplement 3*). However, in more complex situations that require spatial or temporal summation across synaptic locations, these properties will have interesting consequences (*Harvey and Svoboda, 2007*). Therefore, future work investigating such mechanisms using modifications of the CRACM technique to look at postsynaptic properties (*Druckmann et al., 2014*; *Little and Carter, 2012*; *MacAskill et al., 2014*; *MacAskill et al., 2012*) is an important future direction.

Overall we have defined a novel circuit that allows BA input to define the activity of parallel output pathways from vH to control motivated behaviour. The anatomical and functional specificity of this circuit provides an ideal substrate upon which to control reward and value-based learning and decision-making, and helps to explain the multiple and varied roles attributed to this circuit.

# Materials and methods

**Key resources table**

| Reagent type (species) or resource | Designation | Source or reference | Identifiers | Additional information |
|---|---|---|---|---|
| Genetic reagent (*Mus musculus*) | *Slc32a1*(VGAT)-IRES-Cre (*vGAT-cre*) | Jackson Laboratory | Stock #016962; RRID:IMSR_JAX:016962 | |
| Genetic reagent (*M. musculus*) | Ai14(RCL-tdT)-D (reporter mice) | Jackson Laboratory | Stock #007914; RRID:IMSR_JAX: 007914 | |
| Genetic reagent (virus) | AAV2/1-CaMKII-GFP | Addgene | Stock #64545-AAV1 | A gift from Edward Boyden |
| Genetic reagent (virus) | AAV2retro-CAG-Cre | UNC vector core (*Tervo et al., 2016*) | | |
| Genetic reagent (virus) | AAV2/1-EF1a-FLEX-hChR2 (H134R)-EYFP | Addgene | Stock #20298-AAV1 | A gift from Karl Deisseroth |
| Genetic reagent (virus) | AAV2/1-hSyn-hChR2 (H134R)-EYFP | Addgene | Stock #26973-AAV1 | A gift from Karl Deisseroth |
| Genetic reagent (virus) | AAV2/1-CaMKII-hChR2 (H134R)-EYFP | Addgene | Stock #26969-AAV1 | A gift from Karl Deisseroth |
| Genetic reagent (virus) | pAAV2/8-hSyn-dF-HA-KORD-IRES-mCitrine | Addgene | Stock #6541-AAV8 | A gift from Bryan Roth |
| Genetic reagent (virus) | AAV2/1.CAG.FLEX.Ruby2sm-Flag.WPRE | Addgene | Stock #98928-AAV1 | A gift from Loren Looger |
| Genetic reagent (virus) | AAV2/9-mDlx-NLS-mRuby2 | Addgene | Stock #99130-AAV1 | A gift from Viviana Gradinaru |
| Genetic reagent (virus) | pAAV2/1-Ef1a-DIO mCherry | Addgene | Stock #114471-AAV1 | A gift from Karl Deisseroth |
| Antibody | Anti-somatostatin antibody, clone YC7 (monoclonal) | Merck Millipore | MAB354; RRID:AB_2255365 | IHC (1:500) |
| Chemical compound, drug | Salvinorin B (SalB) | Hello Bio | HB4887 | |
| Chemical compound, drug | Cholera Toxin Subunit B (recombinant), Alexa Fluor 647 Conjugate | Thermo Fisher Scientific | C34778 | |
| Chemical compound, drug | Cholera Toxin Subunit B (recombinant), Alexa Fluor 594 Conjugate | Thermo Fisher Scientific | C34777 | |
| Chemical compound, drug | Cholera Toxin Subunit B (recombinant), Alexa Fluor 488 Conjugate | Thermo Fisher Scientific | C34775 | |
| Software, algorithm | Python 3.7 | https://www.python.org/ | RRID:SCR_008394 | |
| Software, algorithm | Jupyter Notebook | https://www.jupyter.org/ | RRID:SCR_018315 | |
| Software, algorithm | ImageJ (Fiji) | https://www.fiji.sc/ | RRID:SCR_002285 | |

## Animals

6–10-week-old (adult) male C57bl/6J mice provided by Charles River were used except where noted. To target inhibitory neurons, we used the *Slc32a1*(VGAT)-IRES-Cre (#016962) knock-in line. To visualise vGAT neurons, we utilised and crossed the vGAT-cre line with Ai14(RCL-tdT)-D reporter mice (#007914), both obtained from Jackson Laboratory and bred in-house. For the vGAT-based

experiments in *Figure 4*, both male and female mice were used and were randomly assigned to experimental groups; numbers of each sex are itemised in the supplemental statistics table. Mice were housed in cages of 2–4 and kept in a humidity- and temperature-controlled environment under a 12 hr light/dark cycle (lights on 7 am to 7 pm) with ad libitum access to food and water. All experiments were approved by the U.K. Home Office as defined by the Animals (Scientific Procedures) Act and University College London ethical guidelines.

## Stereotaxic surgery
### Retrograde tracers
Red and green fluorescent retrobeads (Lumafluor, Inc) for electrophysiological recordings.

Cholera toxin subunit B (CTXβ) tagged with Alexa 555, 488 or 647 (Molecular Probes) for histology experiments.

## Viruses

AAV2/1-CaMKII-GFP (a gift from Edward Boyden; Addgene #64545)
AAV2retro-CAG-Cre (UNC vector core)
AAV2/1-EF1a-FLEX-hChR2(H134R)-EYFP (a gift from Karl Deisseroth; Addgene #20298-AAV1)
AAV2/1-hSyn-hChR2(H134R)-EYFP (a gift from Karl Deisseroth; Addgene #26973-AAV1)
AAV2/1-CaMKII-hChR2(H134R)-EYFP (a gift from Karl Deisseroth; Addgene #26969-AAV1)
pAAV2/8-hSyn-dF-HA-KORD-IRES-mCitrine (a gift from Bryan Roth; Addgene #6541-AAV8)
AAV2/1.CAG.FLEX.Ruby2sm-Flag.WPRE (a gift from Loren Looger; Addgene #98928-AAV1)
AAV2/9-mDlx-NLS-mRuby2 (a gift from Viviana Gradinaru; Addgene #99130-AAV1)
pAAV2/1-Ef1a-DIO mCherry (a gift from Karl Deisseroth; Addgene 114471-AAV1)

## Surgery
Stereotaxic injections were performed on 7–10-week-old mice anaesthetised with isoflurane (4% induction, 1–2% maintenance) and injections carried out as previously described (*Sanchez-Bellot and MacAskill, 2021*; *Wee and MacAskill, 2020*). Briefly, the skull was exposed with a single incision and small holes drilled in the skull directly above the injection site. Injections were carried out using long-shaft borosilicate glass pipettes with a tip diameter of ~10–50 µm. Pipettes were back-filled with mineral oil and front-filled with ~0.8 µl of the substance to be injected. A total volume of 250–300 nl of each virus was injected at each location in ~14 or 28 nl increments every 30 s. If two or more substances were injected in the same region, they were mixed prior to injection. The pipette was left in place for an additional 10–15 min to minimise diffusion and then slowly removed. If optic fibres were also implanted, these were inserted immediately after virus injection, secured with 1–2 skull screws and cemented in place with C&B superbond. Injection coordinates were as follows (mm relative to bregma):

Infralimbic PFC: ML: ± 0.4; RC: + 2.3; DV: - 2.4
NAc: ML: ± 0.9, RC: + 1.1; DV: - 4.6
BA: ML: ± 3.4, RC: - 1.7; DV: - 4.8
vH: ML: ± 3.2, RC: - 3.7; DV: - 4.5

After injection, the wound was sutured and sealed, and mice recovered for ~30 min on a heat pad before they were returned to their home cage. Animals received carprofen in their drinking water (0.05 mg/ml) for 48 hr post-surgery as well as subcutaneously during surgery (0.5 mg/kg). Expression occurred in the injected brain region for ~2 weeks for WT animals and ~4 weeks for vGAT animals until behavioural testing, preparation of acute slices for physiology experiments or fixation for histology. The locations of injection sites were verified for each experiment.

## Anatomy
### Histology
Mice were perfused with 4% PFA (wt/vol) in PBS, pH 7.4, and the brains dissected and postfixed overnight at 4°C as previously described (*MacAskill et al., 2014*; *Sanchez-Bellot and MacAskill, 2021*; *Wee and MacAskill, 2020*). 70-µm-thick slices were cut using a vibratome (Campden Instruments) in either the transverse or coronal planes as described in the figure legends. For immunostaining,

slices were incubated for 3 hr in blocking solution to avoid non-specific protein binding. The blocking solution contained 3% bovine serum albumin, 0.5% Triton and phosphate buffer solution. Slices were incubated at 4°C in blocking solution with either 1:200 SOM antibody (MAB354, Millipore) or 1:5000 anti-DDDDK tag (anti-FLAG Tag to label smFP, ab1258, Abcam). Incubation was overnight for FLAG staining and for 48 hr for SOM staining. Slices were washed three times with PBS for 5–20 min at room temperature. Slices were then incubated for a minimum of 3 hr at room temperature with appropriate secondary antibodies and washed three times with PBS for 15–20 min before they were mounted. Slices were mounted on Superfrost glass slides with ProLong Gold or ProLong Glass (for visualisation of GFP) antifade mounting medium (Molecular Probes). NucBlue was included to label gross anatomy. Imaging was carried out with a Zeiss Axio Scan Z1 using standard filter sets for excitation/emission at 365-445/50 nm, 470/40-525/50 nm, 545/25-605/70 nm and 640/30-690/50 nm. Raw images were analysed with Fiji.

## Whole-brain registration

Cell counting of cholera-labelled inputs was conducted using WholeBrain (*Fürth et al., 2018*; *Wee and MacAskill, 2020*). After acquiring the imaged sections and exporting them as 16-bit depth image files, images were manually assigned a bregma coordinate (AP –6.0 to 0.0 mm) and processed using WholeBrain (*Fürth et al., 2018*) and custom cell counting routines written in R (*Wee and MacAskill, 2020*). The workflow comprised (1) segmentation of cells and brain section, (2) registration of the cells to the ABA and (3) analysis of anatomically registered cells. As tissue section damage impairs the automatic registration implemented on the WholeBrain platform, sections with poor registration were manually registered to the atlas plate using corresponding points to clear anatomical landmarks. Once all cells had been registered, the cell counts were further manually filtered from the dataset to remove false-positive cells (e.g. debris).

Each cell registered to a brain region was classified as belonging to an anatomically defined region as defined by the ABA brain structure ontology. Information on the ABA hierarchical ontology was scraped from the ABA API (http://api.brain-map.org/api/v2/structure_graph_download/1.json) using custom Python routines. For quantification of input fractions, cells residing in different layers within the same structure, for example, COAa1, COAa2, etc., were agglomerated across layers and subdivisions and counted as residing in one single region (e.g. COAa). Structures included as part of BA were 'BLAa', 'BLAv', 'BLAp', 'BMAa', 'BMAp', 'LA', 'COAa', 'COApl', 'COApm', 'MEAa', 'MEAav', 'MEApd', 'MEApv', 'CEAc', 'CEAm', 'CEAl', 'PAA', 'PA'. For co-localisation of VGAT+ and CTXβ-labelled neurons, images acquired as above were manually annotated with single- and dual-labelled neurons using Napari (napari contributors, 2019, doi.10.5281/zenodo.3555620). Whole-brain distributions were visualised using the Brainrender package for Python (*Claudi et al., 2020*).

## Electrophysiology

### Slice preparation

Hippocampal recordings were studied in acute transverse slices. Mice were anaesthetised with a lethal dose of ketamine and xylazine, and perfused intracardially with ice-cold external solution containing (in mM) 190 sucrose, 25 glucose, 10 NaCl, 25 $NaHCO_3$, 1.2 $NaH_2PO_4$, 2.5 KCl, 1 $Na^+$ ascorbate, 2 $Na^+$ pyruvate, 7 $MgCl_2$ and 0.5 $CaCl_2$, bubbled with 95% $O_2$ and 5% $CO_2$. Slices (400 µm thick) were cut in this solution and then transferred to artificial cerebrospinal fluid (aCSF) containing (in mM) 125 NaCl, 22.5 glucose, 25 $NaHCO_3$, 1.25 $NaH_2PO_4$, 2.5 KCl, 1 $Na^+$ ascorbate, 3 $Na^+$ pyruvate, 1 $MgCl_2$ and 2 $CaCl_2$, bubbled with 95% $O_2$ and 5% $CO_2$. After 30 min at 35°C, slices were stored for 30 min at 24°C. All experiments were conducted at room temperature (22–24°C). All chemicals were from Sigma, Hello Bio or Tocris.

## Whole-cell electrophysiology

Whole-cell recordings were made from hippocampal pyramidal neurons retrogradely labelled with retrobeads which were identified by their fluorescent cell bodies and targeted with Dodt contrast microscopy, as previously described (*MacAskill et al., 2014*; *Sanchez-Bellot and MacAskill, 2021*; *Wee and MacAskill, 2020*). For sequential paired recordings, neurons were identified within a single field of view at the same depth into the slice. The recording order was counterbalanced to avoid any potential complications that could be associated with rundown. For current-clamp recordings,

borosilicate recording pipettes (4–6 MΩ) were filled with (in mM) 135 K-gluconate, 10 HEPES, 7 KCl, 10 Na-phosphocreatine, 10 EGTA, 4 MgATP and 0.4 NaGTP. For voltage-clamp experiments, three internals were used, First, in *Figures 2, 4 and 5I–P*, a Cs-gluconate-based internal was used containing (in mM) 135 gluconic acid, 10 HEPES, 7 KCl, 10 Na-phosphocreatine, 4 MgATP, 0.4 NaGTP, 10 TEA and 2 QX-314. Excitatory and inhibitory currents were electrically isolated by setting the holding potential at –70 mV (excitation) and 0 mV (inhibition) and recording in the presence of APV. Experiments in *Figure 5A, B, E-H* were carried out using current-clamp internal in APV in order to carry out post-stimulation analysis of intrinsic properties of recorded interneurons. Finally, to record inhibitory currents at –70 mV in *Figure 5C and D* we used a high chloride internal (in mM): 135 CsCl, 10 HEPES, 7 KCl, 10 Na-phosphocreatine, 10 EGTA, 4 MgATP, 0.3 NaGTP, 10 TEA and 2 QX-314. Recordings were made using a Multiclamp 700B amplifier, with electrical signals filtered at 4 kHz and sampled at 10 kHz.

Presynaptic glutamate release was triggered by illuminating ChR2 in the presynaptic terminals of long-range inputs into the slice, as previously described (*Sanchez-Bellot and MacAskill, 2021*; *Wee and MacAskill, 2020*). Wide-field illumination was achieved via a ×40 objective with brief pulses of blue light from an LED centred at 473 nm (CoolLED pE-4000/Thorlabs M470L4-C1, with appropriate excitation-emission filters). Light intensity was measured as 4–7 mW at the back aperture of the objective and was constant between all cell pairs.

## Electrophysiology data acquisition and analysis
Electrophysiology data were acquired using National Instruments boards and WinWCP (University of Strathclyde). Optical stimulation was via wide-field irradiance with 473 nm LED light (CoolLED) as described above. Data were analysed using custom routines written in Python 3.6, imported using the neo package in Python (*Garcia et al., 2014*). For connectivity analysis, a cell was considered connected if the average of light-induced response was greater than 2 standard deviations above baseline. Amplitudes of responses were calculated as the average of a 2 ms window around the peak of the response. Current step data (Figure S2) were analysed using routines based around the eFEL package in Python (Blue Brain Project).

## Integrate-and-fire model
An integrate-and-fire model was constructed using the Brian2 package in Python (*Stimberg et al., 2019*). 1000 vH-BA, vH-NAc and vH-PFC neurons were modelled interspersed with 80 interneurons (*Lee et al., 2014a*). Neurons were set to have a leak conductance, resting potential, spike threshold and membrane capacitance based on the literature and our current-clamp recordings (*Figure 3*): leak conductance 5.5 nS; resting potential –70 mV, spiking threshold –35 mV, membrane capacitance 200 pF. Connectivity of the local vH circuit was based on our electrophysiology recordings. AMPA receptor connections were 1 nS and were modelled with a tau of 5 ms. GABA receptor-mediated connections were 3 nS and modelled with a tau of 10 ms. Feedback connectivity from each pyramidal neuron population was connected at a probability of 0.1. The probability of connection of local interneurons to pyramidal neurons was based on *Figure 5* and was 0.8 for vH-NAc neurons and 0.4 for vH-BA and vH-PFC neurons, each with a 3 nS GABA conductance. To simulate excitatory BA input, neurons were supplied with 50,000 BA inputs timed as a Poisson distribution with an average rate of 10 Hz. Each neuron was connected to this input with a probability of 0.1, where the strength of the synaptic connection was randomly drawn from a normal distribution defined by our electrophysiology experiments in *Figure 4* (vH-BA 0.3 ± 0.2 nS, vH-NAc 0.3 ± 0.2 nS, vH-PFC 0.03 ± 0.2 nS, interneurons 0.3 ± 0.2 nS). To simulate BA inhibitory input, neurons were again supplied with 50,000 BA inputs timed as a Poisson distribution with an average rate of 10 Hz, but the connection probability was calculated as a proportion of excitatory input and varied across runs. As before, the strength of each synaptic connection was randomly drawn from a normal distribution defined by our electrophysiology experiments in *Figure 4* (vH-BA 0.3 ± 0.2 nS, vH-NAc 0.08 ± 0.2 nS, vH-PFC 0.03 ± 0.2 nS, interneurons 0.3 ± 0.2 nS). Each simulation was run five times at each level of inhibitory connection strength, with the length of simulation 500 ms for each run. To investigate the influence of feedforward and feedback connection probability, proportion of overlap between populations and postsynaptic mechanism (*Figure 6—figure supplements 1–3*), we systematically altered these parameters for each run. Model output was analysed as total spikes produced by each neuronal population over the course of 500 ms.

## Behaviour

After sufficient time for surgical recovery and viral expression (>4 weeks), mice underwent multiple rounds of habituation. Mice were habituated to the behavioural testing area in their home cage for 30 min prior to testing each day. Mice were habituated to handling for at least 3 days, followed by 1–2 days of habituation to the optical tether in their home cage for 10 min.

## Real-time place preference

Axon terminals were labelled as described above, and a 200 µm optical fibre was implanted unilaterally 100 µm above the stimulation area (vH). After habituation (above), behaviour was assessed using an RTPP task. On day 1, mice were exposed to the three-chamber arena (24 cm × 16 cm × 30 cm) for 15 min without stimulation to allow habituation and also to ensure no large side bias was present. The testing chamber was made out of black acrylic, was symmetrical and had no odour, visual or tactile cues to distinguish either side of the arena. The arena was thoroughly wiped down with 70% ethanol between each trial. Mice were excluded if they spent more than 80% of their time in one side of the chamber during this habitation session. On day 2, 20 Hz light stimulation was delivered via a 473 nm laser, coupled to a patch cord (7–10 mW at the end of the patch cord) to activate ChR2-positive terminals. Real-time light delivery was based on the location of the mouse in the RTPP apparatus, where light stimulation occurred only when the mouse was in the light-paired side of the arena. The paired side was chosen randomly for each mouse and each session, thus in combination with the lack of explicit cues in the chamber, this assay represents acute place preference and not learned preference over sessions. Time spent in the light-paired and control side of the arena over the course of the 15 min session was scored for each mouse using automated tracking analysis (Bonsai). For experiments involving pharmacogenetics (*Figures 7 and 8*), mice first underwent habituation and laser-only trials as before, and data from control animals were used to replicate the original RTPP cohort (Figures 6A–C and 7A–C). Next, mice were given 1–2 daily s.c. injections of 100 µl DMSO (10% in saline) for habituation, before undergoing two further days of testing – first with DMSO as a control and with 10 mg/kg SalB the next day to avoid any spillover effects of the SalB injection. All injections were given 15 min prior to RTPP session. Control mice for optogenetics expressed GFP in BA. Control mice for KORD experiments consisted of a mixture of mice expressing smFP in vH$^{NAc}$ neurons and mice lacking expression in vH, all of which received an injection of both DMSO and SalB. No differences were seen across the two conditions, and so data were pooled. Several mice were removed from the analysis due to missed injections (seven), broken implants (six), evidence of light-induced seizure (five – all of which were subsequently found to have bleed of ChR2 into vH) and due to an early error in SalB administration (three). No group was overrepresented in any of these issues.

## Statistics

Summary data are reported throughout the figures either as boxplots, which show the median, 75th and 95th percentile as bar, box and whiskers, respectively, or as line plots showing mean ± SEM. Example physiology and imaging traces are represented as the median ± SEM across experiments. Data were assessed using statistical tests described in the supplementary statistics summary, utilising the Pingouin statistical package for Python (*Vallat, 2018*). Significance was defined as p<0.05, all tests were two-sided. No statistical test was run to determine sample size a priori. The sample sizes we chose are similar to those used in previous publications. Animals were randomly assigned to a virus cohort (e.g. ChR2 versus GFP), and where possible the experimenter was blinded to each mouse's virus assignment when the experiment was performed. This was sometimes not possible due to, for example, the presence of the injection site in the recorded slice.

## Acknowledgements

We thank members of the MacAskill Laboratory for helpful comments on the manuscript. AFM was supported by a Sir Henry Dale Fellowship jointly funded by the Wellcome Trust and the Royal Society (grant number 109360/Z/15/Z) and by a UCL Excellence Fellowship. RA was supported by a King Fahad Medical City Studentship. RWSW was supported by a UCL Graduate Research Scholarship and a UCL Overseas Research Scholarship. KM and JP were supported by the Wellcome Trust 4 year PhD in Neuroscience at UCL (grant numbers 215165/Z/18/Z and 222292/Z/20/Z, respectively).

# Additional information

## Funding

| Funder | Grant reference number | Author |
|---|---|---|
| Wellcome Trust | 109360/Z/15/Z | Andrew F MacAskill |
| Wellcome Trust | 215165/Z/18/Z | Karyna Mishchanchuk |
| Wellcome Trust | 222292/Z/20/Z | Jessica Passlack |
| King Fahad Medical City | | Rawan AlSubaie |

The funders had no role in study design, data collection and interpretation, or the decision to submit the work for publication.

## Author contributions

Rawan AlSubaie, Conceptualization, Formal analysis, Investigation, Writing – review and editing; Ryan WS Wee, Anne Ritoux, Karyna Mishchanchuk, Jessica Passlack, Daniel Regester, Investigation; Andrew F MacAskill, Conceptualization, Formal analysis, Funding acquisition, Investigation, Resources, Supervision, Writing – original draft, Writing – review and editing

## Author ORCIDs

Rawan AlSubaie ⓘ http://orcid.org/0000-0002-0721-4744
Ryan WS Wee ⓘ http://orcid.org/0000-0003-0273-5521
Anne Ritoux ⓘ http://orcid.org/0000-0002-5760-6172
Karyna Mishchanchuk ⓘ http://orcid.org/0000-0002-0996-790X
Jessica Passlack ⓘ http://orcid.org/0000-0002-1043-3980
Daniel Regester ⓘ http://orcid.org/0000-0002-5372-7479
Andrew F MacAskill ⓘ http://orcid.org/0000-0002-0196-3779

## Ethics

All experiments were approved by the U.K. Home Office as defined by the Animals (Scientific Procedures) Act, and University College London ethical guidelines.

## Decision letter and Author response

Decision letter https://doi.org/10.7554/eLife.74758.sa1
Author response https://doi.org/10.7554/eLife.74758.sa2

# Additional files

## Supplementary files

- Supplementary file 1. Statistical summary of main figures.
- Supplementary file 2. Statistical summary of figure supplements.
- Transparent reporting form

## Data availability

All main figures are supplied with source data used to generate the figures.

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
