## [Editor Report]

This manuscript represents an important piece of work that defines the cellular basis of hippocampal-amygdala functional connectivity in rodents.

---

## [Decision Letter]

**Decision letter after peer review:**

[Editors’ note: the authors submitted for reconsideration following the decision after peer review. What follows is the decision letter after the first round of review.]

Thank you for submitting your work entitled "Control of Parallel Hippocampal Output Pathways by Amygdalar Long-Range Inhibition" for consideration by *eLife*. Your article has been reviewed by 3 peer reviewers, one of whom is a member of our Board of Reviewing Editors, and the evaluation has been overseen by a Senior Editor. The following individual involved in review of your submission has agreed to reveal their identity: Sadegh Nabavi (Reviewer #3).

We are sorry to say that, after consultation with the reviewers, we have decided that this version of the work will not be considered further for publication by *eLife*.

That said, there was considerable enthusiasm for your manuscript and its potential to report impactful findings. Thus, we would be willing to consider a revised version of your manuscript as a new submission, if you are able to well address each of the concerns noted below. If you choose to resubmit your manuscript to *eLife*, it will be editorially evaluated prior to being sent to review. We will endeavor to send a revision to the same reviewers, but cannot guarantee this. The following are the main points that must be addressed.

1) Circuit mapping is sub-optimal. This key issue has been raised by all the three reviewers and should be carefully addressed. For example, the Authors should provide quantitative EPSCs data based on objective criteria, as explained by comment #1 of reviewer 1.

2) Key and novel information on the identity of the GABAergic projections from the BA and their cellular targets in the VH should be given.

3) Divergent VH CA1 projections should be detected, their inputs investigated and added as new experimental and modelling data.

4) Most of the behavioural experiments are underpowered and need to be improved by substantial increase in the sample size that would also make the statistical comparisons more meaningful.

5) The authors should provide high quality evidence of the tracer-virus injection sites, in order to evaluate the potential viral spread, and the fiber projection patterns.

6) Additional technical improvements should be made and submitted in a revised version. They include using more physiological intracellular calcium concentration in some in vitro experiments, as well as to provide a better control for the chemogenetic experiments.

*Reviewer #1:*

A strength of the manuscript is that it includes data obtained with a variety of complementary approaches, such as CRACM, electrophysiology, behavior and modeling that all contribute toward a comprehensive definition of BA-VH connectivity.

However, there are weaknesses that make the claims raised not as direct as they should. Furthermore, the data submitted does contains some important gaps. Specifically: CRACM should significantly improve, a better characterization of the "novel" BA long-range GABAergic neurons should be provided, the identification of the VH cellular targets (CA1, CA3 pyramidal cells?) should be provided too, VH pyramidal cells with multiple projection areas should be investigated.

I found that the quality and the logic of the submitted manuscript (text, figures, data analysis, methods, and discussion) is quite good. One exception is that some key references are not cited (please see below). Regarding the data weaknesses, I have the specific major comments.

1) CRACM is sub-optimal. The present data interpretation relies on intrinsic assumptions without experimental support. It is important to consider that the first paper that used CRACM (Petreanu et al., Nature Neurosci 2007) contained important controls and methodological considerations that should not be forgotten. For example, the size of evoked EPSC = (a) number of connected axons x (b) amplitude of unitary EPSP. In the absence of known (a) and (b) is really quite hard to come up with a quantitative comparison of EPSC values obtained from the activation of different inputs from different transfected animals, (d) control data to verify the level of ChR-2 expression in the various transfected inputs.

The Authors should consider of re-working their approach and provide at least: estimate the amplitude of unitary EPSP for each connection examined, laser power versus spiking probably and latency plots for each neuron type transfected with ChR2, EPSC data normalization based on an estimation of the number of presynaptic fibers transfected for each input.

2) The identity of the BA GABAergic projection neuron should be provided as part of this manuscript. To include this neuron type characterization (somastostatin-expressing GABAergic cell?) would render this manuscript significantly stronger.

3) The cellular target identity of the BA projection to the VH also remains unknown. From the present data, it is not clear whether the BA inputs target the CA1 and/or CA3 VH.

4) Ciocchi et al; Science, 2015 reported VH CA1 pyramidal neurons with divergent projections to the amygdala, medial prefrontal cortex and nucleus accumbens with specific behavioral roles. The Author should detect these divergent VH CA1 pyramidal neurons and investigate their role in modeling and behavior.

*Reviewer #2:*

The work described in this manuscript very elegantly explores the functional connectivity of BA projections to vHPC neurons, using an original and skilful combination of optogenetics and retrograde tracing approaches, as well as the potential role of these projections in goal-directed behavior. Overall, the manuscript benefits from extensive studies on the functional connectivity between BA and vHPC, and contributes important novel information to the field including solid evidence for the long postulated long-range inhibitory efferents from the BA.

My enthusiasm for the work is, however, diminished by the preliminary nature of the behavioural experiments and the lack of cogent experimental evidence for some of the claims, including the validity of the ratio used to compare the strength of the functional BA-inputs to different populations of vHPC principal neurons shown in Figure 3 and Figure 4.

1) I am not sure of the validity of the ratio used to compare the strength of the functional BA-inputs to different populations of vHPC principal neurons shown in Figure 3 and Figure 4. The authors do not provide well-defined and objective criteria. These ratio could be, therefore, strongly biased. This raises questions about the soundness of the claim that feedforward inhibition is markedly skewed towards nAcb-projecting neurons. It also seems that the purported differential impact of feedforward inhibition activated by excitatory BA inputs onto BA-projecting vs nAcb-projecting vHPC neurons largely depends on a single data point. (Figure 4 panel P). The choice of the statistical test (Wilcoxon Rank test), based on the assumption that the recordings can be considered matched samples, has probably favored a type 1 statistical error.

2) Most of the behavioural experiments appear largely underpowered. If we consider the intrinsic variability of the place preference test (also substantiated by the data presented by the authors in Figure 6 and 7), the number of animals used (and in particular for the test animals, i.e. SalB-treated) is, in my opinion, far too small. It is fair to say that being in most cases paired tests, the n can be smaller. Despite this, an n of 4 is definitely suboptimal, even if the apparent statistical effect is clear (e.g. Figure 6 panel K).

3) Given the relative complexity of the design of the behavioural experiments (particularly those combining opto- and chemogenetics) involving the use of multiple viral vectors and injection sites, I wonder how many animals had to be excluded and based on which criteria. This information should be provided. Since the injection sites are provided (Figure 6 suppl 2 and Figure 7 suppl 2), I assume that the viral spread was analysed. A representative image of the injection side should be provided alongside with the projection patterns in vHPC for each of the key experiments.

4) The authors have completely neglected to discuss and consider in their interpretation of their results the fact that vHPC principal neurons often have more than one long-range projection, e.g. mPFC-projecting neurons can also project to the nAcb (Ciocchi et al; Science, 2015). How can they reconcile their finding that BA-inputs target nAcb-projecting neurons but not mPFC-projecting ones with the data published by Ciocchi et al?*Reviewer #3:*

This work provides a detailed map of mono-synaptic connections between the BA and the vH. Particularly valuable, they use CRACM to identify the inhibitory and excitatory mono-synaptic connections. They extend their investigation of mono-synaptic connections to the vH output neurons (BA, NAc, and mPFC projecting neurons). Then they build an integrate-and-fire network model, constrained by their experimental data. Finally, they test the model's prediction at the behavioral level.

Overall, this work is carefully designed and nicely executed. Although, I am not qualified in evaluating their modeling, I liked their approach. It is a well-rounded experimental design, where they use their own set of data to construct a model with predictive power that later put in test. That is, they bridged the gap between slice electrophysiology and behavior with circuit modeling.

However, some of the main claims require more experimental evidence. This includes increasing the power for the behavioral experiments, evaluating a potential contribution of topographical bias within the BA, verifying that high concentration of the calcium chelator is not having unintended consequences, and a more thorough validation of the effect of SalB on action potentials.

– In this work CRACM was used to quantify the projection biases between two different regions. The way it is done, however, ignores the topographical biases that exist within the source region. For example, figure 3D-F clearly shows a stark difference between the optically evoked responses in vHmPFC and vHBA neurons. This may indicate, as favored by the authors, a low connectivity between the BA and the vHmPFC, or, alternatively, a topographical bias within the BA, where different parts preferentially target vHmPFC and vHBA neurons.

– In real time place preference part, some of the manipulations were done with as few as 4 mice (ex. Figure 6K. 7C). The number of mice used in 6I is more reassuring.

– All the recordings were done with an unusually high concentration of the calcium chelator EGTA (10 mM). EGTA is more mobile compared to the endogenous calcium buffers, and this could enhance the diffusion of calcium and result in unintended consequences. For example, the high concentration of EGTA may reduce the basal level of intracellular calcium concentration which in turn may change the basal synaptic transmission. The authors should run a control showing the concentration of EGTA used here does not affect the basal transmission. This can be done with a 15-minute long paired recording of the optically-evoked responses from the excitatory neurons in the vH, with one cell filled with high EGTA (10 mM) and the other with low EGTA (0.5 mM).

– SalB injection produces a modest hyperpolarization (2mV) (figure 6F). Whether this is sufficient to block action potential is not convincingly tested. The number of action potentials should be presented at more depolarized states (figure 6E).

---

## [Author Response]

[Editors’ note: the authors resubmitted a revised version of the paper for consideration. What follows is the authors’ response to the first round of review.]

The following are the main points that must be addressed.

We thank the reviewers for their enthusiasm for the manuscript. We hope our additional experiments, analysis and discussion itemised below will mitigate their concerns.

1) Circuit mapping is sub-optimal. This key issue has been raised by all the three reviewers and should be carefully addressed. For example, the Authors should provide quantitative EPSCs data based on objective criteria, as explained by comment #1 of reviewer 1.

We thank the reviewers for their comments, and are very keen to make sure that our circuit mapping successfully addresses the questions we are asking in the paper. Overall, however, we think there is some confusion that stems from us not adequately explaining the rationale and details of how the experiments were carried out. We apologise for this lack of clarity, and have substantially updated the text throughout he manuscript to more thoroughly explain our experimental setup. We summarise our rationale below:

The reviewers – and particularly reviewer one – rightly pointed out important limitations to the CRACM technique. These are excellent comments, with which we very much agree. In particular, a key issue with CRACM is the difficulty in reconciling data from different experiments, with variability in e.g. injection and expression levels of ChR2 meaning that light evoked post synaptic responses will be highly variable from slice to slice. This makes a quantitative comparison of input across slices extremely challenging without detailed histology, indirect quantification of axon density and ChR2 expression level, and the use of optical quantal stimulation – which is challenging due to the non-physiological nature of ChR2 currents in presynaptic terminals.

However, as reviewer one describes, Petreanu et al. 2007 very nicely characterised this technique, and as a result it has become the basis of a large number of studies, including from my lab and many others. Importantly for our study, in Figure 4 of Petreanu 2007, the authors developed a means to circumvent many of the caveats associated with CRACM and allow quantifiable comparisons across cell types. In short, by comparing relative input across neighbouring neurons within the same slice. This method has subsequently become the gold standard in the field, and although it is much harder experimentally – involving recording from 2 neighbouring projection-defined neurons in the same slice and field of view – allows the direct comparison of synaptic input across cell types, without relying on the indirect estimations described above.

In our study we make sure to use this method throughout our comparisons to mitigate these potential caveats of CRACM. For example, in Figure 4D-F of our original manuscript, for each experiment we recorded one neuron projecting to BA, and one neuron projecting to PFC in the same slice and same field of view. This way the expression and activation of presynaptic ChR2 is identical for both post-synaptic neurons, removing the experimental caveats mentioned by the reviewers. This is why all of our comparisons (and accompanying statistics) are presented as ratios rather than absolute values – the objective measure is the relative input onto each pair of recorded neurons. Overall, our results in Figure 4D-F show that across different experiments, each BA projecting neuron receives a greater proportion of excitatory input than a neighbouring PFC-projecting neuron when the same presynaptic terminals are excited.

Importantly there are limited instances where this approach is experimentally not possible (e.g. Figure 5A-H). In this case, we have explicitly stated this, and have only inferred binary connectivity, and not any quantitative differences in connectivity across connections. To complement these experiments, we then investigated the potential consequences of differential connectivity in our model (Figure 6 —figure supplement 1).

Overall, we apologise for the lack of clarity in our original description of our experiments. We have now substantially updated the results starting on p 6, line 36, and p 10, line 15 to make the reasoning behind our approach clear and have added a substantial section to the discussion starting on p19, line 8 outlining the limitations of the CRACM technique, and how these can be mitigated by carrying out experiments in the way they have been carried out in our study. We have also specifically addressed each of the reviewers’ points in detail below. Together, we hope that this clarification of the rationale underpinning our approach will satisfy the reviewers concerns.

2) Key and novel information on the identity of the GABAergic projections from the BA and their cellular targets in the VH should be given.

Identity of BA inhibitory projections:

We agree with the reviewers that this is a very interesting point. Numerous studies (reviewed in McDonald and Mott 2016, and discussed in our original manuscript on p15-16, now p 18 in the new manuscript) have previously carried out extensive anatomical investigation of long-range inhibitory projections into hippocampus. These include from PFC, Entorhinal Cortex, BA, and septum. Interestingly across all of these experiments, a common factor is the molecular identity of the inhibitory projections often involves somatostatin expression. Indeed, it has previously been shown using retrograde tracing that the long-range inhibitory input from amygdala to the hippocampal formation most likely arises from somatostatin positive, and parvalbumin negative neurons.

However, we agree with the reviewers that it would be a helpful addition to show this is also the case in our hands, and so we have repeated these experiments. We injected fluorescently conjugated choleratoxin into the vH and compared overlap of retrogradely labelled neurons in BA with somatostatin immunostaining. We found that a proportion of retrogradely labelled neurons were positive for somatostatin. This is consistent with previous work, and is now presented in Figure 2 —figure supplement 2 and described on p 4, line 35, and discussed on p 18, line 28.

Identity of vH neurons:

We apologise that this was not made clearer in our original manuscript. The target area of BA input we focussed on was the CA1 / subiculum border – the area where the vast majority of long-range output projections arise from (as described on p2, p5, p6, and p15 of the original manuscript, with example images and physiological characterisation of the neurons presented in Figure 3 —figure supplement 1). We have now substantially updated the text to highlight the identity of our recording sites more clearly on p 5, line 12, and have moved our characterisation of CA1 / subiculum output populations to the main Figure 3. We have also carried out anatomical analysis of axon projection patterns from both excitatory and inhibitory injections in BA, showing innervation of BA input in this region, now presented in Figure 2 —figure supplement 3, and described on p6, line 16.

3) Divergent VH CA1 projections should be detected, their inputs investigated and added as new experimental and modelling data.

We thank the reviewers for pointing this out. Multiple studies, including detailed studies from our own lab (e.g. Wee and MacAskill, 2020, Naber and Witter 1998) have shown that a small proportion of neurons in ventral hippocampus project to multiple downstream regions. However, in the CA1/subiculum border where we are recording, it is the general consensus that only small percentages (ranging from 2 – 10 %) of neurons are dual projecting. Due to the very sparse nature of these double projecting neurons it is not feasible to carry out the technically challenging paired recording technique to record from these small subpopulations. However, to address the reviewer’s concerns, we have carried out a number of extra experiments, and added substantial extra discussion:

a) We have performed additional, double retrograde labelling experiments that allows us to quantify the proportion of neurons that project to more than one downstream region at the CA1 / subiculum border where we are recording. Consistent with multiple previous studies we found this to be a small proportion – between 2-20%. This is now presented in Figure 3D and described on p6, line 32.

b) We have carried out a number of new in silico experiments using our integrate and fire model to investigate the circuit consequences of having these subpopulations follow each combination of potential connectivity rules predicted by our experiments. In particular, we were interested in the possibility that retrogradely labelled neurons recorded in our dataset may in fact project to more than one downstream area. If this was the case, then a proportion of our recordings from e.g. PFC projecting neurons may also project to NAc. In this scenario, increased collateralisation would result in increased overlap in the properties of each population of neurons.

Therefore, we systematically increased the % of neurons with dual projections from 0% (the same as our original model) to 60 % (where 60% of the neurons in the network had properties from all three populations, and only 40 % had the properties we recorded in our physiology experiments). Interestingly, while we found that altering both local connectivity, and BA input markedly altered the influence of BA input on the network, the switch from BA and NAc projecting activity with increasing long-range inhibition remained intact, even at extreme levels where 40 – 60 % of neurons had synaptic properties defined as if they projected to multiple downstream areas. Therefore, our modelling suggests that the local and long-range connectivity rules are robust to collateralisation, and that this robustness allows for up to ten times higher collateralisation than found in our retrograde labelling experiments. This is now presented in Figure 6 —figure supplement 2 and described on p 12, line 5.

c) Finally, we very much agree with the reviewers that the contrast between Stephane Ciocchi’s work finding large overlap of projection populations, with the large anatomy literature suggesting these neurons only have very small overlap. We have therefore added a substantial section to the discussion to address this. In particular we have discussed the idea that the dual projection recordings in the Ciocchi paper may be from a specific part of proximal CA1 that has only relatively sparse long range projection neurons (around 5-10% of total neurons). These neurons, specifically in proximal CA1, have previously been shown to have a higher collateralisation (up to 30%, Naber and Witter). In contrast, our study focussed on the area of ventral hippocampus containing the bulk of long-range output neurons, located substantially more distally in CA1, at the border between CA1 and subiculum. We think this may be the cause of the discrepancy. In addition, we have also discussed some practical issues that should be taken into account when comparing retrograde tracing such as efficiency of labelling. Finally, we made sure to point out that a comparison of BA innervation across the proximal – distal axis of CA1 and subiculum would be a very interesting future direction. This is especially interesting considering the increasing evidence for a distinction of function along the proximal distal axis of the hippocampus. This extra discussion can be found starting on p 17, line 22.

4) Most of the behavioural experiments are underpowered and need to be improved by substantial increase in the sample size that would also make the statistical comparisons more meaningful.

We have carried out behavioural testing on a second cohort of mice for both KORD experiments in Figures 6 and 7 to increase the n to a minimum of 7 per group. We hope that these additional experiments will satisfy the reviewers.

5) The authors should provide high quality evidence of the tracer-virus injection sites, in order to evaluate the potential viral spread, and the fiber projection patterns.

We thank the reviewer for pointing this out. We have provided example histology images, axon tracing data for both excitatory and inhibitory input from BA into vH, and multiple example injection sites. This is provided in Figure 2 —figure supplement 3 and 5, Figure 3B, and Figure 7—figure supplement 2. We have also verified this axon tracing using tracing data from the Allen connectivity database, which is provided as Figure 2 —figure supplement 4. We hope this will be suitable to address the reviewers’ concerns. We also hope that this will help address reviewer one’s concern over the exact location of our recordings (point 2 above).

6) Additional technical improvements should be made and submitted in a revised version. They include using more physiological intracellular calcium concentration in some in vitro experiments, as well as to provide a better control for the chemogenetic experiments.

EGTA: Our voltage clamp internal contained QX-314 to block sodium channels, TEA and Cs to block potassium channels, and EGTA to buffer calcium. Our goal with the addition of each of these chemicals was to increase the accuracy of our voltage clamp recordings by removing potentially contaminating currents. Therefore, our internal solution is intentionally non physiological, as this helps with the interpretation of our experiments. 10 mM EGTA is often used in voltage clamp experiments where holding voltage is shifted between -70 and 0 mV as in our study (e.g. Rigby et al. 2015, Bats et al., 2012). In addition, as all of our recordings are investigating the relative input into a pair of neighbouring neurons in the same slice, any potential effect of this extra buffering would be equal for both of the recorded neurons, and so would be internally controlled.

Combined with our more detailed description of our internally controlled paired recording experimental setup (see point one above), we hope our explanation of the reasoning behind utilising EGTA in our internal will satisfy the reviewers concerns. In addition, we have added a section to the discussion on p 18, line 40 pointing out that an investigation of plasticity in this projection would be an important future direction.

KORD controls: The extra controls requested by reviewer 3 were present in the original manuscript, in Figure 6, figure supplement 1C – now Figure 7, figure supplement 1C. We have now explicitly referenced these experiments in the manuscript on p 12, line 34.

Reviewer #1:I found that the quality and the logic of the submitted manuscript (text, figures, data analysis, methods, and discussion) is quite good. One exception is that some key references are not cited (please see below). Regarding the data weaknesses, I have the specific major comments.1) CRACM is sub-optimal. The present data interpretation relies on intrinsic assumptions without experimental support. It is important to consider that the first paper that used CRACM (Petreanu et al., Nature Neurosci 2007) contained important controls and methodological considerations that should not be forgotten. For example, the size of evoked EPSC = (a) number of connected axons x (b) amplitude of unitary EPSP. In the absence of known (a) and (b) is really quite hard to come up with a quantitative comparison of EPSC values obtained from the activation of different inputs from different transfected animals, (d) control data to verify the level of ChR-2 expression in the various transfected inputs.The Authors should consider of re-working their approach and provide at least: estimate the amplitude of unitary EPSP for each connection examined, laser power versus spiking probably and latency plots for each neuron type transfected with ChR2, EPSC data normalization based on an estimation of the number of presynaptic fibers transfected for each input.

We have added a large section to the manuscript outlining the reasoning behind the design of these experiments, which are summarised in point 1 above. However, to further mitigate the reviewers concerns we have added further explanation and performed new experiments as detailed below:

a) 'the Authors should consider of re-working their approach and provide laser power versus spiking probably (and latency plots) for each neuron type transfected with ChR2 since this type of measurement will provide a rationale for using a certain light intensity for each set of stimulation’.

The purpose of our study was to investigate long range input from amygdala into the different projection populations in ventral hippocampus. As this is a long-range projection, this is only possible using ChR2-assisted circuit mapping (or CRACM), which allows the direct stimulation of axons within the slice containing hippocampal neurons, despite the fact that these axons are severed from the soma during slicing. Therefore, while this is an excellent suggestion for local connectivity studies, recording light-evoked spiking in separate slices containing the soma of ChR2 infected cells in the amygdala will not be informative with respect to the ability of light to directly stimulate amygdalar axons in slices of hippocampus. As we described in our main response, instead we only compare pairs of recorded neurons while stimulating the same axon terminals. Thus, postsynaptic responses are internally controlled within each pair. We have added this to the results on p 6, line 36, and p 10, line 15 to make the reasoning behind our approach clear and have added a substantial section to the discussion starting on p19, line 8, discussing the potential caveats with the CRACM approach and how they might eb mitigated.

b) 'This technique may be also give a clue of the light stimulation needed to activate a single presynaptic fiber that would be an ideal protocol.’ [and that the authors should attempt to carry out], 'EPSC data normalization based on an estimation of the number of presynaptic fibers transfected for each input' … ’the size of evoked EPSC = (a) number of connected axons x (b) amplitude of unitary EPSP. In the absence of known (a) and (b) is really quite hard to come up with a quantitative comparison of EPSC values obtained from the activation of different inputs from different transfected animals’.

As stated above, the purpose of our study was to investigate long range input from amygdala into ventral hippocampus, and particularly the relative influence of this input on the different projection populations in hippocampus. Therefore, our goal was to use a method that allowed as controlled a comparison as possible of input across two populations of cells. As mentioned above this is very important to us, especially – as pointed out by the reviewer – due to the inherent variability of the CRACM approach due to differences in for example the expression level of ChR2, or the exact location of injection or recording.

Because of this, and as discussed in detail in above in point 1, in our study we make sure to mitigate the potential caveats of CRACM as much as possible, and for each experiment we recorded one neuron from each population of interest in the same slice and field of view. This way the expression and activation of presynaptic ChR2 is identical for both post-synaptic neurons, removing the experimental caveats mentioned by the reviewer.

In contrast, the technique described by the reviewer aims to ask a very different question, which is to ask what the unitary synaptic properties of the connections from amygdala onto hippocampal neurons are. This is a fascinating area, and one which we have personally investigated in depth in other studies (Macaskill et al. 2012, MacAskill et al. 2014). However, it is a different question to the one addressed in the paper. Crucially, as the reviewer themself points out, it would be extremely difficult to accurately compare total input onto two populations of neurons using this technique, and it would rely on indirect estimations based on histology, cell or axon counting across animals and optical-quantal analysis. This is in contrast to the direct, within-experiment comparison gained with the method we use, and which has been used repeatedly in the field (three examples from the last year: Anastasiades et al. 2021, *Neuron*; Young… and Petreanu 2021, *eLife*; Martinez-Garcia et al. 2020, *Nature*). Overall, we want to highlight that for the question asked in our paper, we think the way the experiment was originally performed in the study was most likely the most appropriate.

However, despite these concerns, we have carried out extra modelling experiments to further investigate this idea. Using our model we explicitly asked if there were potential functional consequences of the specific mechanism underlying the difference in relative input across the populations. We built different versions of our model where the differences in BA input onto each population are due to either unitary postsynaptic amplitude, or probability of connection. We found that -as anticipated in our simple system – irrespective of the underlying reason for differences in total synaptic input, the functional consequences for the vH circuit remained the same. This is presented in Figure 6 —figure supplement 3, and discussed on p 12, line 5.

In addition, we have added substantially to our discussion on p 19, line 26, to address the interesting consequences of each of the mechanisms on circuit function. For example, in situations that require temporal or spatial summation of inputs across the dendrites. Together we think this improves the discussion, and we hope it will satisfy the reviewers concerns.

2) The identity of the BA GABAergic projection neuron should be provided as part of this manuscript. To include this neuron type characterization (somastostatin-expressing GABAergic cell?) would render this manuscript significantly stronger.

We agree with the reviewer and as noted above have now carried out extra experiments to address this point. Numerous studies (reviewed in McDonald and Mott 2016, and discussed in our original manuscript on p15-16) have previously shown that the long-range inhibitory input to the hippocampal formation most likely arises from somatostatin positive, and parvalbumin negative neurons from across the amygdala. To confirm this is the case in our hands, we have carried out retrograde anatomy experiments combined with immunostaining for somatostatin, and show that somatostatin positive neurons in amygdala project to hippocampus. This is presented in Figure 2 —figure supplement 2, and on p 4, line 35, and discussed on p 18, line 28.

3) The cellular target identity of the BA projection to the VH also remains unknown. From the present data, it is not clear whether the BA inputs target the CA1 and/or CA3 VH.

We apologise that this was not made clearer. The target area of BA input we focussed on was the CA1 / subiculum border – the area where the vast majority of long-range output projections arise from (as described on p2, p5, p6, and p15 of the original manuscript, with example images and physiological characterisation of the neurons presented in Figure 3 —figure supplement 1). We have now substantially updated the text to highlight the identity of our recording sites more clearly on p 5, line 12, and have moved our characterisation of CA1 / subiculum output populations to the main Figure 3. We have also carried out anatomical analysis of axon projection patterns from both excitatory and inhibitory injections in BA, showing innervation of BA input in this region, now presented in Figure 2 —figure supplement 3 and 4, and described on p6, line 16.

4) Ciocchi et al; Science, 2015 reported VH CA1 pyramidal neurons with divergent projections to the amygdala, medial prefrontal cortex and nucleus accumbens with specific behavioral roles. The Author should detect these divergent VH CA1 pyramidal neurons and investigate their role in modeling and behavior.

We agree with the reviewer that this is an important point, and indeed the majority of our manuscript (Figure 3 – Figure 8) is concerned with understanding how BA input differentially impacts each of these unique projection populations, and what the potential behavioural consequences of this targeting are.

We also agree with the reviewer, as well as reviewer 2 and 3 that the function and connectivity of the relatively small proportion of projection neurons with collaterals to more than one downstream region may be distinct and have unique functional roles. As outlined in the response to point 3 we have performed a number of new experiments and added substantial discussion to address this. We hope this addresses the reviewer’s concern.

Reviewer #2:1) I am not sure of the validity of the ratio used to compare the strength of the functional BA-inputs to different populations of vHPC principal neurons shown in Figure 3 and Figure 4. The authors do not provide well-defined and objective criteria. These ratio could be, therefore, strongly biased. This raises questions about the soundness of the claim that feedforward inhibition is markedly skewed towards nAcb-projecting neurons. It also seems that the purported differential impact of feedforward inhibition activated by excitatory BA inputs onto BA-projecting vs nAcb-projecting vHPC neurons largely depends on a single data point. (Figure 4 panel P). The choice of the statistical test (Wilcoxon Rank test), based on the assumption that the recordings can be considered matched samples, has probably favored a type 1 statistical error.

We thank the reviewer for their comment and apologise that our rationale for our experiments was not clear in the original submission. As stated in our response to point 1, our data is presented as a ratio because each data point represents the light evoked response in a paired recording between two neurons projecting to distinct downstream regions, while keeping presynaptic stimulation constant. This practically and technically challenging experimental setup was designed to mitigate concerns about variability in ChR2-evoked synaptic responses due to injection site location, recording location, depth in the slice, and ChR2 expression. Therefore, we hope by better explaining our rationale for using paired comparisons this mitigates the reviewer’s concerns. In short, taking Figure 5 as an example, each data point shows that for the response recorded in a vH^BA^ neuron, the relative response of a neighbouring VH^NAc^ neuron in the same slice to exactly the same stimulus is consistently higher. We have now substantially updated the results starting on p 6, line 36, and p 10, line 15 to make the reasoning behind our approach clear and have added a substantial section to the discussion starting on p19, line 8 outlining the limitations of the CRACM technique, and how these can be mitigated by carrying out experiments in the way they have been carried out in our study.

2) Most of the behavioural experiments appear largely underpowered. If we consider the intrinsic variability of the place preference test (also substantiated by the data presented by the authors in Figure 6 and 7), the number of animals used (and in particular for the test animals, i.e. SalB-treated) is, in my opinion, far too small. It is fair to say that being in most cases paired tests, the n can be smaller. Despite this, an n of 4 is definitely suboptimal, even if the apparent statistical effect is clear (e.g. Figure 6 panel K).

We agree with the reviewer and have now carried out a new cohort of mice for each of the KORD experimental groups in Figure 7 and 8, which allowed us to increase the *n* to at least 7 for each condition. We hope this is now satisfactory.

3) Given the relative complexity of the design of the behavioural experiments (particularly those combining opto- and chemogenetics) involving the use of multiple viral vectors and injection sites, I wonder how many animals had to be excluded and based on which criteria. This information should be provided. Since the injection sites are provided (Figure 6 suppl 2 and Figure 7 suppl 2), I assume that the viral spread was analysed. A representative image of the injection side should be provided alongside with the projection patterns in vHPC for each of the key experiments.

We agree with the reviewer and regret not including this information in the original manuscript. Indeed, a number of animals were excluded – including due to mistargeting of the injections or fibres, and an error in administering SalB. This is now explicitly stated with numbers in each case in the methods p 24, line 35.

4) The authors have completely neglected to discuss and consider in their interpretation of their results the fact that vHPC principal neurons often have more than one long-range projection, e.g. mPFC-projecting neurons can also project to the nAcb (Ciocchi et al; Science, 2015). How can they reconcile their finding that BA-inputs target nAcb-projecting neurons but not mPFC-projecting ones with the data published by Ciocchi et al?

We agree with the reviewer, as well as reviewer 1 and 3 that the function and connectivity of the relatively small proportion of projection neurons with collaterals to more than one downstream region may be distinct and have unique functional roles. As outlined in the response to point 3 we have performed a number of new experiments and added substantial discussion to address this. We hope this addresses the reviewer’s concern.

Reviewer #3:– In this work CRACM was used to quantify the projection biases between two different regions. The way it is done, however, ignores the topographical biases that exist within the source region. For example, figure 3D-F clearly shows a stark difference between the optically evoked responses in vHmPFC and vHBA neurons. This may indicate, as favored by the authors, a low connectivity between the BA and the vHmPFC, or, alternatively, a topographical bias within the BA, where different parts preferentially target vHmPFC and vHBA neurons.

We thank the reviewer for pointing this out, and apologise that our original description of our experimental design was not clear. As described in point 1 above, the aim of our study was to investigate the relative synaptic input from each BA input onto the different vH output populations. Therefore, for each experiment we carried out paired recordings – we recorded from a BA projecting neuron as well as a neighbouring NAc or PFC projecting neuron in the same slice. All of our comparisons are carried out solely across this internally controlled experiment. Therefore, in new Figure 4D-F (previously Figure 3D-F referred to by the reviewer) our results show that there is greater input onto BA- projecting neurons compared to neighbouring PFC -projecting neurons in the same slice, and the same topographical position, with the same viral injection, and stimulating the same axon terminals. We took pains to carry out our experiments in this difficult fashion in order to control for the large variability in the CRACM technique, as described above in detail by reviewer 1, and also by us in the response to point 1. As discussed above we have now substantially updated the results starting on p 6, line 36, and p 10, line 15 to make the reasoning behind our approach clear, and have added a substantial section to the discussion starting on p19, line 8 outlining the limitations of the CRACM technique, and how these can be mitigated by carrying out experiments in the way they have been carried out in our study.

Therefore, overall, our results therefore show this targeting is present despite the potential topographical variability suggested by the reviewer. We agree that the topographical distribution of connectivity is extremely interesting, and have directly investigated this in previous studies (e.g. Wee and MacAskill 2020). We have now added a section to the discussion on p 17, line 34 where we explicitly state it will be key to investigate how BAvH connectivity varies across vH topography in the future.

In addition, we agree that a very interesting future direction will be to investigate how unique nuclei in BA connect with each projection population in vH, and how this connectivity varies along each axis of the hippocampus. We have now added a new section to the discussion on p 16, line 22 highlighting this important future direction, where we note this will most likely require the use of intersectional genetic targeting.

– In real time place preference part, some of the manipulations were done with as few as 4 mice (ex. Figure 6K. 7C). The number of mice used in 6I is more reassuring.

We agree with the reviewer, and have now carried out a new cohort of mice for each of the KORD experimental groups in Figure 7 and 8, which allowed us to increase the *n* to at least 7 for each condition. We hope this is now satisfactory.

– All the recordings were done with an unusually high concentration of the calcium chelator EGTA (10 mM). EGTA is more mobile compared to the endogenous calcium buffers, and this could enhance the diffusion of calcium and result in unintended consequences. For example, the high concentration of EGTA may reduce the basal level of intracellular calcium concentration which in turn may change the basal synaptic transmission. The authors should run a control showing the concentration of EGTA used here does not affect the basal transmission. This can be done with a 15-minute long paired recording of the optically-evoked responses from the excitatory neurons in the vH, with one cell filled with high EGTA (10 mM) and the other with low EGTA (0.5 mM).

Our voltage clamp internal contained QX-314 to block sodium channels, TEA and Cs to block potassium channels, and EGTA to buffer calcium. Our goal with the addition of each of these chemicals was to increase the accuracy of our voltage clamp recordings by removing potentially contaminating currents. Therefore, our internal solution is intentionally non physiological, as this helps with the interpretation of our experiments. 10 mM EGTA is often used in voltage clamp experiments where holding voltage is shifted between -70 and 0 mV as in our study (e.g. Rigby et al. 2015, Bats et al., 2012). In addition, as described in detail above, all of our recordings are investigating the relative input into a pair of neighbouring neurons in the same slice, any potential effect of this extra buffering would be equal for both of the recorded neurons, and so would be internally controlled.

Combined with our more detailed description of our internally controlled paired recording experimental setup (see point 1 above), we hope our explanation of the reasoning behind utilising EGTA in our internal will satisfy the reviewers concerns.

However, we do agree that future work should investigate the physiological consequences, and potential for plasticity in this pathway. We have now added a new section to the discussion to highlight this important point on p 18, line 40.

– SalB injection produces a modest hyperpolarization (2mV) (figure 6F). Whether this is sufficient to block action potential is not convincingly tested. The number of action potentials should be presented at more depolarized states (figure 6E).

We apologise that our previous manuscript did not more clearly describe our control experiments. These experiments were present in the original manuscript, in Figure 6 —figure supplement 1C (now Figure 7 —figure supplement 1C). We have now updated the text to explicitly reference these experiments more robustly on p 12, line 34. We hope that this will satisfy the reviewers concerns.